# RESIDUAL CONNECTIONS HARM GENERATIVE REPRESENTATION LEARNING

## ABSTRACT

We show that introducing a weighting factor to reduce the influence of identity shortcuts in residual networks significantly enhances semantic feature learning in generative representation learning frameworks, such as masked autoencoders (MAEs) and diffusion models. Our modification improves linear probing accuracy for both, notably increasing ImageNet accuracy from 67.8% to 72.7% for MAEs with a VIT-B/16 backbone, while also boosting generation quality for diffusion models. This significant gap suggests that, while residual connection structure serves an essential role in facilitating gradient propagation, it may have a harmful side effect of reducing capacity for abstract learning by virtue of injecting an echo of shallower representations into deeper layers. We ameliorate this downside via a fixed formula for monotonically decreasing the contribution of identity connections as layer depth increases. Our design promotes the gradual development of feature abstractions, without impacting network trainability. Analyzing the representations learned by our modified residual networks, we find correlation between low effective feature rank and downstream task performance.

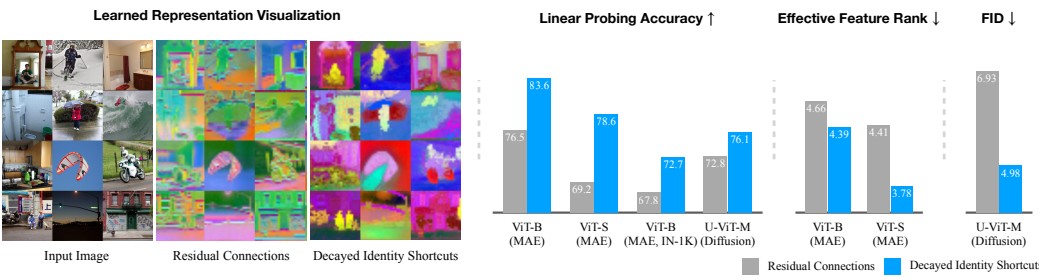

Figure 1: We design *decayed identity shortcuts* (Figure 2), a variant of residual connections, to facilitate self-supervised representation learning. Compared to standard residual connections, our approach yields superior abstract semantic features (*left*, visualized using Zhang et al. (2024)'s approach), whose leading components pop out object instances and classes. Quantitative evaluation shows our architecture encourages lower feature rank and achieves a substantial increase in linear probing accuracy for both MAE and diffusion models, along with enhanced generation quality for diffusion models (*right*). These improvements require no additional learnable parameters.

## 1 INTRODUCTION

Residual networks (ResNets) (He et al., 2016) define a connection structure that has achieved near-universal adoption into modern architectures for deep learning. At the time of their development, supervised learning (*e.g.,* ImageNet (Deng et al., 2009) classification) was the driving force behind the evolution of convolutional neural network (CNN) architectures. Residual networks solved a key issue: CNNs constructed of more than approximately 20 convolutional layers in sequence became difficult to train, leading to shallower networks outperforming deeper ones, unless additional techniques, such as auxiliary outputs (Szegedy et al., 2015) or batch normalization (Ioffe & Szegedy, 2015), were employed. Both ResNets, and their predecessor, highway networks (Srivastava et al., 2015a) provide elegant solutions to this trainability problem by endowing the network architecture with

Figure 2: Our *decayed identity shortcuts* introduce a depth-dependent scaling factor to shortcuts in a residual network, thereby modulating the contribution of preceding layers and fostering greater abstraction in deeper layers. A simple schema for controlling decay factor $\alpha$ suffices to improve feature learning in both MAEs and diffusion models, as well as diffusion model generation quality.

alternative shortcut pathways along which to propagate gradients. Highway networks present a more general formulation that modulates these shortcut connections with learned gating functions. However, given their sufficient empirical effectiveness, the simplicity of ResNet's identity shortcuts (residual connections) makes them a preferred technique.

While solving the gradient propagation issue, residual connections impose a specific functional form on the network; between residual connections, each layer (or block of layers) learns to produce an update slated to be added to its own input. This incremental functional form may influence the computational procedures learned by the network (Greff et al., 2017). Alternatives to residual and highway networks exist that do not share this functional form, but implement other kinds of skip-connection scaffolding in order to assist gradient propagation (Larsson et al., 2017; Huang et al., 2017; Zhu et al., 2018). Thus, shortcut pathways, rather than a specific form of skip connection, are the essential ingredient to enable the training of very deep networks. Nevertheless, nearly all modern large-scale models, including those based on the transformer architecture (Vaswani et al., 2017) incorporate the standard identity shortcut residual connection.

This design choice holds, even as deep learning has shifted into an era driven by self-supervised training. The shift to self-supervision brings to the forefront new learning paradigms, including those based on contrastive (Wu et al., 2018; He et al., 2020; Chen et al., 2020; Caron et al., 2021; Grill et al., 2020), generative (Goodfellow et al., 2014; Karras et al., 2021; Ho et al., 2020; Song et al., 2021a; 2023; Rombach et al., 2022), and autoencoding (Kingma & Welling, 2013; He et al., 2022; Li et al., 2023) objectives. Many systems in the generative and autoencoding paradigms rely on "encoder-decoder" architectures, often styled after the original U-Net (Ronneberger et al., 2015), which contains additional long-range shortcuts between corresponding layers in mirrored symmetry about a central bottleneck. With representation learning as a goal, one typically desires that the middle bottleneck layer produce a feature embedding reflecting abstract semantic properties. The interaction of skip-connection scaffolding for gradient propagation with encoder-decoder architectures, self-supervised training objectives, and bottleneck representations has not been carefully reconsidered. This is a worrisome oversight, especially since, even in the supervised setting with standard classification architectures, prior work suggests that unweighted identity shortcuts may be a suboptimal design decision (Savarese & Figueiredo, 2017; Fischer et al., 2023).

Intuitively, identity shortcut connections may not be entirely appropriate for capturing high-level, semantic representations as they directly inject low-level, high-frequency details of inputs into outputs, potentially compromising feature abstraction. We explore this issue within generative learning frameworks, including masked autoencoders (MAEs) (He et al., 2022) and diffusion models (Ho et al., 2020), leading paradigms for self-supervised image representation learning and generation. Our experiments demonstrate that identity shortcuts significantly harm semantic feature learning in comparison to an alternative we propose: gradually decay the weight of the identity shortcut over the depth of the network, thereby reducing information flow through it (Figure 2). With increasing layer depth, our approach facilitates a smooth transition from a residual to a feed-forward architecture, while maintaining sufficient connectivity to train the network effectively. Unlike prior work on

learned gating (Srivastava et al., 2015a) or reweighting (Savarese & Figueiredo, 2017) mechanisms for residual connections, our method is a forced decay scheme governed by a single hyperparameter.

A parallel motivation for our design stems from Huh et al. (2021), who show that features from residual blocks have higher rank than those produced by comparative feed-forward blocks. The smooth transition between residual and feed-forward behavior induced by our decay scheme regularizes deeper features toward exhibiting low-rank characteristics. Section 6 experimentally explores the correlation between our decayed identity shortcuts and low-rank feature representations. Figure 1 previews the corresponding improvements to representation learning. Our contributions are:

- We introduce decayed identity shortcuts, a simple architectural mechanism which enhances semantic feature abstraction in masked autoencoders and diffusion models.

- We identify a key correlation between our decayed identity shortcuts and low-rank inductive bias, empirically validating that our method improves classification accuracy and yields low-rank features with distinct cluster structures.

- Our novel design within an MAE yields a substantial performance boost in linear probing on ImageNet-1K (Deng et al., 2009) (72.7% from a baseline 67.8%).

- In diffusion models, our design improves both feature learning and generation quality.

- Ablation studies on ImageNet-100 show that smaller models equipped with decayed identity shortcuts outperform larger ones using standard residual connections. A VIT-S/16 model (Dosovitskiy et al., 2021) with our shortcut design outperforms a baseline VIT-B/16 (78.5% *vs.* 76.5%).

## 2 RELATED WORK

**Self-supervised representation learning.** Recent advancements (Achiam et al., 2023; Kirillov et al., 2023; Rombach et al., 2022; Team et al., 2023; Shi et al., 2020; Ramesh et al., 2021) in deep learning follow a common scaling law, in which a model's performance consistently improves with its capacity and the size of the training data. This effect can be observed in large language models (LLMs), which are trained on vast amounts of internet text, enabling them to perform some tasks at human level (Laskar et al., 2023) and exhibit remarkable zero-shot capabilities (Kojima et al., 2022). These models are trained using next-token-prediction, allowing them to be trained without labeled data. In contrast, the progress of this scaling law in computer vision has largely depended on annotated data. For instance, the Segment Anything model (Kirillov et al., 2023) leverages 1 billion human-annotated masks, and state-of-the-art image generators (Ramesh et al., 2021) require training on huge datasets of text-image pairs (Schuhmann et al., 2022). However, the vast volume of unlabeled visual data and desire for continued scaling motivates a transition to self-supervised learning.

At present, two families of approaches to self-supervised visual representation learning appear particularly promising: contrastive learning (Wu et al., 2018; He et al., 2020; Chen et al., 2020; Caron et al., 2021; Grill et al., 2020), which trains a discriminative model to maximize mutual information across image augmentations, and generative learning, via masked image modeling (Bao et al., 2022; He et al., 2022; Chen et al., 2024), which trains to reconstruct occluded pixels, or via diffusion denoising (Song et al., 2021b; Ho et al., 2020; Song et al., 2021a), which trains to reverse a process that mixes images with Gausssian noise. Some hybrid approaches (Zhou et al., 2022; Huang et al., 2023; Li et al., 2023) combine both families. Despite advancements, neither has demonstrated the same scalability (Singh et al., 2023) as seen in LLMs. This challenge is additional motivation for reconsidering the foundations of self-supervised network architectures.

**Residual and skip-connection architectures.** Highway networks (Greff et al., 2017) first propose an additive skip connection structure to provide a scaffolding for gradient propagation when training very deep (*e.g.,* 100 layer) networks. Motivated by the gating mechanisms within LSTMs (Hochreiter & Schmidhuber, 1997), this solution uses learned gating functions to weight each combination of identity and layer output branches. Residual networks (He et al., 2016) are a simplification that removes these learned coefficients. DenseNet (Huang et al., 2017) and FractalNet (Larsson et al., 2017) demonstrate that access to gradient paths of multiple lengths are the core requirement of training scaffolding, by introducing skip-connection structures with other functional forms. DenseNet utilizes feature concatenation instead of addition, while FractalNet imposes a recursive tree-like architecture combining subnetworks of multiple depths.

Zhu et al. (2018) explore variants of ResNets and DenseNets with fewer points of combination between different internal paths, demonstrating that a sparser scaffolding structure may be more robust as network depth increases to thousands of layers. Savarese & Figueiredo (2017) add a scalar gating functional to the layer output in residual networks, yielding a hybrid design between residual and highway networks; learning this scalar gating provides a consistent benefit to classification accuracy. Fischer et al. (2023) develop a weighting scheme for residual connections based upon a sensitivity analysis of signal propagation within a ResNet. To date, none of these potential improvements have seen broad adoption.

**Low rank bias in neural networks.** Over-parameterized neural networks exhibit surprising generalization capabilities, a finding seemingly in contradiction with classical machine learning theory (Neyshabur et al., 2019). This phenomenon implies the existence of some form of implicit regularization that prevents the model from overfitting. From the perspective of neural network parameterizations, Arora et al. (2019) suggest that linear models with more layers tend to converge to minimal norm solutions. In the context of CNNs, Huh et al. (2021) demonstrates that stacking more feed-forward layers compels the model to seek solutions of a lower rank, and Jing et al. (2020) reinforce this finding by adding more layers to an autoencoder's bottleneck, thereby creating a representation bottleneck. In vision transformers, Geshkovski et al. (2024) examine the connection between attention blocks and mean-shift clustering (Cheng, 1995), showing that repeated attention operations result in low-rank outputs. Moreover, Dong et al. (2021) reveal that eliminating the shortcut connection from residual attention blocks causes features to degenerate to rank 1 structures doubly exponentially. From a different perspective, recent work (Radhakrishnan et al., 2022; Beaglehole et al., 2023; Radhakrishnan et al., 2024) shows training algorithms implicitly induce low-rank behavior in neural networks. Radhakrishnan et al. (2024) study the dimensionality reduction behavior of a recursive feature machine (Radhakrishnan et al., 2022) and effectively verify performance on low-rank matrix recovery.

## 3 METHOD

Prior works show that deeper feed-forward architectures have an inductive bias towards producing low-rank feature maps, while ResNets do not display the same behavior (Huh et al., 2021). However, despite this bias, deeper feed-forward architectures are typically less effective and generalize worse than ResNets (He et al., 2016). We aim to combine the properties of both feed-forward networks and ResNets, using the low-rank prior to enhance the abstraction capability of the network while maintaining the core benefits of the residual block, including stable training and the capacity to construct deeper models.

### 3.1 DECAYED IDENTITY SHORTCUTS

**Feed-forward layers.** Consider a neural network of $L$ layers. For each layer $l$ parameterized with $\boldsymbol{\theta}_l$, the operation of a feed-forward neural network can be described as:

$$\boldsymbol{x}_{l+1} = f_{\boldsymbol{\theta}_l}(\boldsymbol{x}_l), \tag{1}$$

where $\boldsymbol{x}_l \in \mathbb{R}^d$ represents the output from the preceding layer, and $f_{\boldsymbol{\theta}_l}$ denotes the transformation applied at the current layer. Although it is widely known that pure feed-forward architectures are susceptible to vanishing gradients when building deeper models, Huh et al. (2021) demonstrates that feed-forward modules offer implicit structural regularization, enabling deep models to generate abstract representations at bottlenecks.

**Residual connections.** To address the optimization problem of vanishing gradients in deeper neural networks, ResNets (He et al., 2016) construct each layer as a residual function, resulting in a modification to Eq. 1:

$$\boldsymbol{x}_{l+1} = \boldsymbol{x}_l + f_{\boldsymbol{\theta}_l}(\boldsymbol{x}_l). \tag{2}$$

This design builds shortcuts from input to output, allowing gradient magnitude to be preserved regardless of the depth of the model. However, a consequence of this design is that the output stays close to the input in practice (Greff et al., 2017), defeating the need to construct complex transformations over depth. The same phenomenon is also observed in highway networks (Srivastava et al., 2015a), which adopt learnable gates $H_\phi(\boldsymbol{x}) \in [0, 1]^d$ in both the residual and skip branches:

$\boldsymbol{x}_{l+1} = H_\phi(\boldsymbol{x}_l) \cdot \boldsymbol{x}_l + (1 - H_\phi(\boldsymbol{x}_l)) \cdot f_{\boldsymbol{\theta}_l}(\boldsymbol{x}_l)$. Although this flexible design allows the model to build the abstraction level over depths, similar to feedforward networks, Srivastava et al. (2015b) finds $H_\phi \approx 1$ for most units, suggesting the model prefers copying the input.

**Decayed identity shortcuts for unsupervised representation learning.** Setting aside the optimization benefits brought by residual connections, we rethink the role of the residual connections from the viewpoint of representation learning. Abstraction can be viewed as invariance to local changes of input and is crucial to the disentanglement of the feature space (Bengio et al., 2013). Prior work suggests that a shortcut path of residual connections tends to preserve high-frequency fine-grained input information (Greff et al., 2017), resulting in decreased feature abstraction. We hypothesize that this lack of abstraction harms the capability of the model to learn meaningful low-level features and that ensuring an abstract structure in the deeper layers of the neural network will help improve representation learning, especially for unsupervised tasks that often use indirect proxy objectives, such as pixel-wise reconstruction loss. Motivated by this hypothesis, we propose to downweight the contribution from the shortcut path:

$$\boldsymbol{x}_{l+1} = \alpha_l \boldsymbol{x}_l + f_{\boldsymbol{\theta}_l}(\boldsymbol{x}_l), \tag{3}$$

where $\alpha_l \in [0, 1]$ is a rescaling factor to the residual path, controlling the information flow through the skip connection. Fully expanding this relation for a network with $L$ layers indexed from 0 to $L - 1$, we have that:

$$\boldsymbol{x}_L = \left(\prod_{l=0}^{L-1} \alpha_l\right) \boldsymbol{x}_0 + \sum_{l=0}^{L-2} \left(\prod_{i=l+1}^{L-1} \alpha_i\right) f_{\boldsymbol{\theta}_l}(\boldsymbol{x}_l) + f_{\boldsymbol{\theta}_{L-1}}(\boldsymbol{x}_{L-1}). \tag{4}$$

We see that the contribution of the input $\boldsymbol{x}_0$ is scaled by each $\alpha_l \leqslant 1$ while each subsequent network block output $f_{\boldsymbol{\theta}_l}(\boldsymbol{x}_l)$ omits scaling factors up to $\alpha_l$. Hence, the contribution of early features of the network is especially down-weighted, preventing the network from passing fine-grained detailed information to the bottleneck $X_L$.

**Decay schema.** Rather than adopting a naive choice of $\alpha_l$ as a constant across all layers, we choose $\alpha_l$ to be a function parameterized by the layer index $l$, where the contribution from the shortcut path is monotonically decreasing when $l$ increases:

$$\alpha_l = 1 - \delta_\alpha l, \tag{5}$$

where $\delta_\alpha := \frac{(1-\alpha_{\min})}{L}$, $\alpha_{L-1} \equiv \alpha_{\min}$ is a minimum scaling factor applied at the final layer $L - 1$. Our formulation brings two primary benefits. First, $\alpha_l$, as a linear interpolation between 0 and 1, acts as a smooth transition between residual connections and feedforward layers, bringing us the optimization benefits seen in the residual connections, while simultaneously encouraging learning the deeper layers to learn more abstract representations. Second, similar to the naive formulation, our method only introduces one extra hyperparameter $\alpha_{\min}$, which is not data-dependent and does not need to be learned.

## 3.2 IMPLEMENTATION STRATEGY

**Skip connections for autoencoders.** Since our method progressively decays the residual connections over network depth, it encourages the most abstract features to be learned by later layer. However, learning a highly abstract bottleneck is detrimental to the training objectives that aim for pixel-wise reconstruction, as they necessitate the preservation of information across all feature levels. To address this, we incorporate standard skip connections between the encoder and decoder, enabling the encoder to directly pass information from shallow layers to the decoder while learning increasingly abstract representations in the deeper encoder layers.

**Stabilizing training with residual zero initialization.** The model exhibits rapid feature norm growth at the beginning of training for $\alpha_{\min} \leqslant 0.7$. We suspect that the model learns to amplify the output feature norm of $f_{\boldsymbol{\theta}_l}(\boldsymbol{x})$ to counteract the significant decay applied to the residual connection. This growth leads to training instability and negatively impacts training convergence. To address this issue, we follow the implementation of previous works (Ho et al., 2020) and initialize the weights of the final output layer in each $f_{\boldsymbol{\theta}_l}$ to zero instead of using the original Xavier uniform initialization (Glorot & Bengio, 2010). This approach significantly enhances training stability by limiting the rate of feature norm growth and enables us to explore training with even lower values of $\alpha_{\min}$.

| Method | Objective | Augmentation | FT | LP |
|---|---|---|---|---|
| MoCo-v3(Chen et al., 2021) | Contrastive Loss | Full | 83.2 | 76.7 |
| DINO(Caron et al., 2021) | Contrastive Loss | Full | 83.3 | 78.2 |
| Con MIM(Yi et al., 2023) | Contrastive Loss | Affine + Mask | 83.7 | 39.3 |
| ADDP(VIT-L) (Tian et al., 2024) | Feature Loss (VQ) | Affine + Mask | 85.9 | 23.8 |
| Latent MIM(Wei et al., 2024) | Feature Loss | Affine+ Mask | 83.0 | 72.0 |
| Data2Vec(Baevski et al., 2022) | Feature Loss | Full+Mask | 84.2 | 68.0 |
| CAE(Chen et al., 2024) | Recon. + Feature Loss | Affine+Mask | 83.8 | 70.4 |
| I-JEPA(Assran et al., 2023) | Feature Loss | Affine+Mask | - | **72.9** |
| MAE(He et al., 2022) | Recon. Loss | Affine+Mask | 83.6 | 67.8 |
| Ours ($\alpha_{\min} = 0.6$) | Recon. Loss | Affine+Mask | 82.9 | **72.7** |

Table 1: **Accuracy of linear classifier based on pre-trained learned representations on the ImageNet-1K dataset.** We evaluate our learned representation using the standard evaluation protocol: linear probing (LP) and fine-tuning (FT). With our simple modification, we substantially improve the MAE ViT-B/16 baseline for linear probing by encouraging the model to learn an increased abstract features over depth. We achieve competitive performance compared to I-JEPA which uses an explicit feature loss.

# 4 EXPERIMENTS ON MASKED AUTOENCODER (MAE)

For masked autoencoders (MAEs) (He et al., 2022), we replace the residual connections in the encoder's MLP and attention blocks with decayed identity shortcuts. The MAE operates by accepting images with a random subset of pixels masked out and learning to recover the discarded pixels. As Section 3.2 describes, we add skip connections between the encoder and decoder to facilitate learning abstract representations at the bottleneck. Since the original MAE has twice the number of encoder layers as decoder layers, we inject output from every other encoder layer into the corresponding decoder layer. To match spatial dimensions, injected encoder features are combined with learnable masked tokens before channel-wise concatenation. The implementation details for the training and evaluation are shown in Section A. He et al. (2022) show the desired representations appear at the end of encoder; we therefore apply our decaying schema only to the encoder.

## 4.1 REPRESENTATION LEARNING ON IMAGENET-1K

We pre-train MAE on the ImageNet-1K train split (Deng et al., 2009) and follow recent works to evaluate the learned representations using linear probing and end-to-end finetuning, where we added a single linear head over learned representations to predict the image categories

For hyperparameters of pretraining MAE, including the learning rate schedule, total training epochs, and mask ratio, we follow the best settings found in the original paper. Please see the appendix for detailed experimental setups.

We report the results in Table 1, where we show the linear probing performance of various self-supervised methods, which we categorize by their objectives, and data augmentation processes. In the top half of the table, we present methods that employ a contrastive loss. Although these methods produce the best probing accuracies, their success depends on a carefully designed data augmentation process, which may need to be tuned for each different data distribution. In the bottom half, we show several methods based on generative architecture, including ours, which do not rely on contrastive objectives. With the exception of Data2Vec, these methods only use a standard random affine data transformation (with random masking), which need not be distribution-reliant. Among these methods, MAE only uses a pixel-wise loss, I-JEPA , Latent MIM, and CAE use a latent feature alignment loss, and CAE uses both. Our method simply extends MAE by constructing an implicit feature bottleneck and shows significant improvements over the MAE baselines (72.7% *vs.* 67.3%), outperforming Data2Vec, Latent MIM and CAE and giving a probing accuracy competitive with I-JEPA, without needing to use explicit feature alignment.

End-to-end fine-tuning, unlike linear probing which only trains a single linear layer, updates the entire network for image classification. Since the features can shift significantly from their pre-

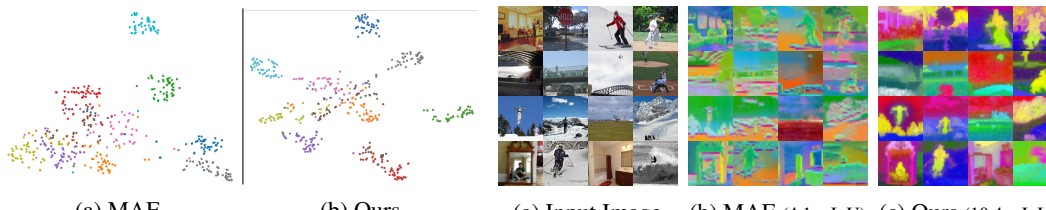

(a) MAE      (b) Ours      (a) Input Image    (b) MAE (4.1 mIoU)    (c) Ours (10.4 mIoU)

Figure 3: **Comparison of t-SNE visualization for models trained with (a) standard residual connection and (b) our method with $\alpha_{\min} = 0.6$.** We visualize the learned embedding using t-SNE on 10 randomly selected categories of ImageNet-100 and we color points within the same category using the same color. Model with standard residual connections (a) have collapsed together while our method (b) forms well structured clusters.

Figure 4: **Visualize learned representations using Zhang et al. (2024).** We project the learned representations onto a 3-channel feature map, visualized as RGB images. Our method learns more abstract and semantically consistent representations compared to the baseline MAE. This visual comparison is further supported by benchmarking on unsupervised semantic segmentation tasks, where our approach achieves better results (10.4 mIoU) compared to the baseline MAE (4.1 mIoU).

training state during end-to-end updating, we argue that this may not accurately reflect the quality of the learned representations. For example, DINO demonstrates superior performance in various downstream vision tasks compared to MAE, but its fine-tuning performance is worse than MAE. Similarly, ConMIM and ADDP exhibit poor linear probing performance, suggesting lower-quality representations, yet their fine-tuning performance surpasses that of contrastive learning methods. Nevertheless, we still provide the fine-tuning results for reference.

### 4.2 EMBDDING ANALYSIS.

We qualitatively evaluate the feature learning in Figure 3 by visualizing the t-SNE embeddings of the learned features for a random subset of test images. The embedding produced from our features displays much clearer separation than baseline, which has struggles to differentiate the categories.

To provide another qualitatively evlauation of our learned representations, we adopt the pixel-wise embedding approaches proposed by Zhang et al. (2024) to group the representations from the last layer of the encoder into a lower dimensional space. We use their default hyperparameters to cluster representations across images of COCO validation set. We visualize in Figure 4 the top 3 eigenvectors for both our approaches with $\alpha_{\min} = 0.7$ and baseline. From the visualization, ours learns abstract representation and the object from the same categories have similar color, indicating a global consistent semantic grouping. The baseline MAE, on the other hand, doesn't show clearly global semantic patterns overall and the low-level representations, e.g., edges, are also high-lightened, indicating a representations from multiple semantic hierarchies are entangled.

We further evaluate the clustering quantitatively, following the postprocessing protocol (Zhang et al., 2024) to produced unsupervised semantic segmentation and report the results as the mean intersection of union (mIoU). Ours (10.41 mIoU) achieve 6.31 mIoU improvement over baseline (4.10 mIoU), which support the qualitatively comparison.

### 4.3 ABLATION STUDIES ON IMAGENET-100

We conduct ablations on several properties of our framework on ImageNet-100. A summary of results can be found in Tables 2 and 3.

**Decay rate $\alpha_{\min}$.** The only parameter of our framework is $\alpha_{\min}$, the minimum scaling factor applied to the identity shortcut at the final layer. In Table 2,we show linear probing scores for varying values of $\alpha_{\min}$. We observe that $\alpha_{\min}$ must be sufficiently small to regularize the flow of information through the residual connection effectively. A reasonably small $\alpha_{\min}$ prevents the deeper layers of the encoder from relying heavily on the residual connections, allowing for more abstract representations in the bottleneck. This yields up to a 7.1% improvement over the ViT-B/16 baseline ($\alpha_{\min} = 1$)

| Backbone \ $\alpha_{\min}$ | 0.5 | 0.6 | 0.7 | 0.8 | 0.9 | 1.0 |
|---|---|---|---|---|---|---|
| ViT-B/16 | 82.3 | **83.6** | 81.8 | 79.8 | 79.2 | 76.5 |
| ViT-S/16 | **78.6** | 78.5 | 78.1 | 75.2 | 73.5 | 69.2 |

Table 2: **Linear Probing Accuracy on ImageNet-100 for our method varying $\alpha_{\min}$ and architecture size.** We conduct ablation studies and demonstrate that linear probing performance for both architectures increases as $\alpha_{\min}$ decreases until around 0.5-0.6. While the larger ViT-B/16 architecture achieves the highest accuracy of 83.6, it is noteworthy that the smaller ViT-S/16, when utilizing our method, outperforms the baseline setting (standard Residual Connection at $\alpha_{\min} = 1$) of ViT-B/16.

| Configurations | UNet | Accuracy |
|---|---|---|
| $\alpha_{\min} = 0.6$ | No | 61.5 |
| $\alpha_{\min} = 0.6$ | Yes | **83.6*** |

| Configurations | Decay Block | $\alpha_{\min}$ | Accuracy |
|---|---|---|---|
| $\boldsymbol{x}_{l+1} = \boldsymbol{x}_l + f_{\boldsymbol{\theta}_l}(\boldsymbol{x}_l)$ | - | — | 76.5 |
| $\boldsymbol{x}_{l+1} = \boldsymbol{x}_l + \sqrt{0.5} f_{\boldsymbol{\theta}_l}(\boldsymbol{x}_l)$ | MLP & Atten. | — | 76.9 |
| $\boldsymbol{x}_{l+1} = \sqrt{0.5}\left(\boldsymbol{x}_l + f_{\boldsymbol{\theta}_l}(\boldsymbol{x}_l)\right)$ | MLP & Atten. | — | 82.6 |
| $\boldsymbol{x}_l = \alpha_l \boldsymbol{x}_l + f_\theta(\boldsymbol{x}_l)$ | Atten. | 0.6 | 79.3 |
| $\boldsymbol{x}_l = \alpha_l \boldsymbol{x}_l + f_\theta(\boldsymbol{x}_l)$ | MLP | 0.6 | 80.6 |
| $\boldsymbol{x}_l = \alpha_l \boldsymbol{x}_l + f_\theta(\boldsymbol{x}_l)$ | MLP & Atten. | 0.6 | **83.6*** |

(a) **Effect of Skip Connections**. Applying the framework without skip connections designed in Section 3.2 results in a severe drop in representation quality.

(b) **Other Decay Schemas**. We conduct ablations using a variety of scalings of the residual connection. We observe that our full method produces the best results.

Table 3: **Linear probing accuracy of ablation experiments using ViT-B/16 on ImageNet-100.** *We duplicate the performance of our $\alpha_{\min} = 0.6$ result from Table 2 for comparison.

in linear probing accuracy on ImageNet-100. On the other hand, if $\alpha_{\min}$ is too small, for example, $\alpha_{\min} \leqslant 0.4$ for ViT-B/16, we observe that the training becomes unstable.

**Architecture size.** In Table 2, we also train both the ViT-B/16 and the smaller ViT-S/16 backbone using varying $\alpha_{\min}$. Our framework is especially effective on the smaller architecture, increasing linear probing performance by 9.4%, compared to the baseline setting (standard Residual Connection at $\alpha_{\min} = 1$). This is consistent with the observation that larger models generalize better (Huh et al., 2021) than smaller models. Our method is able to significantly improve the smaller ViT-S/16 and slightly close the gap between the differently-sized architectures.

**Skip connections.** Another critical design choice in our network is to include skip connections that are not in the original MAE. As discussed in Section 3.2, if the MAE does not use skip connections, the bottleneck layer must preserve all information to reconstruct the input image accurately. This is opposed to learn abstract representations at bottleneck. These contrary effects significantly degrade the representation learned by the model, leading to a 22.1% drop in the linear probing score, as we report in Table 3a.

**Different decay schema.** We also explore decay schema, with results summarized in Table 3b: (1) Scaling both branches of the residual blocks simultaneously by applying a constant factor, $\alpha = \sqrt{0.5}$, to both $\boldsymbol{x}$ and $f_{\boldsymbol{\theta}_l}(\boldsymbol{x})$. (2) Scaling only $f_{\boldsymbol{\theta}_l}$ using the same constant factor, $\alpha = \sqrt{0.5}$. (3) Applying our proposed schema exclusively to either the attention or MLP branch.

Among these, (2) shows no significant improvement over the baseline, while (1) yields some improvement but still underperforms compared to our approach. By analyzing (1) and (2), we demonstrate that the representation gains are due to down-weighting the skip connection branch. Notably, recent diffusion models (Karras et al., 2018; Song et al., 2021b; Karras et al., 2020; 2022) have employed (1) in their designs. However, applying decay only to the MLP or attention branch reduces the overall decaying effect across the network, resulting in lower performance compared to our schema, which achieves the best performance among the tested designs.

## 5 EXPERIMENTS ON DIFFUSION MODELS

**Diffusion models.** We use U-ViT (Bao et al., 2023), a ViT-based diffusion model with skip connections between the encoder and decoder, as the baseline for our diffusion model experiments. Recent

| | $\alpha_{\min}$ | Linear Probing (Acc)↑ | | | | Generation quality (FID)↓ | | | |
|---|---|---|---|---|---|---|---|---|---|
| Dataset | | 1.0 | 0.8 | 0.7 | 0.6 | 1.0 | 0.8 | 0.7 | 0.6 |
| CIFAR-100 (Uncon.) | | 62.47 | 63.58 | **66.86** | 64.63 | 14.34 | 11.65 | **8.99** | 11.71 |
| ImageNet-100 (Uncond.) | | 72.8 | 74.5 | **76.1** | 75.8 | 44.40 | **40.96** | 41.17 | 43.51 |
| ImageNet-100 (Class Cond.) | | - | - | - | - | 6.93 | 5.75 | 5.11 | **4.98** |

Table 4: In diffusion models, we demonstrate that our proposed decayed identity shortcut enhances probing accuracy and improves generation quality across various datasets and configurations.

studies (Yang & Wang, 2023; Baranchuk et al., 2022) suggest that diffusion models learn the best semantic representations near the decoder's latter stages. Therefore, we apply our proposed decay mechanism up to the end of the decoder. While this design might be suboptimal, as the smallest decay factor may not align with the layers holding the best semantic representations, we demonstrate in practice that this simple approach effectively enhances both the learned representations and the quality of generated outputs.

**Experimental details.** We utilize the default scheduler and sampler from U-ViT (Bao et al., 2023), replacing only the residual connections with our proposed decayed shortcut connections. We train unconditional diffusion models on CIFAR-100 and ImageNet-100 without using image class labels. Additionally, we train a class-conditional diffusion model on ImageNet-100 to validate our design across different tasks. For ImageNet-100, instead of training directly on pixels, we adopt a latent diffusion (Rombach et al., 2022) approach by running the model in the latent space of a pretrained VAE, which reduces input resolutions from 256x256x3 to 32x32x4. We use U-ViT-Mid for ImageNet-100 and U-ViT-small for CIFAR-100. For model and training details, please refer to Bao et al. (2023).

We evaluate the learned representations with linear probing and we train a linear classifier over the frozen representations. We report the results as the best configurations, including the choices of layer index and noise level, that yields the best performance

**Results.** Our results are presented in Table 4, where we demonstrate that replacing residual connections with our proposed decayed identity shortcuts consistently enhances representation quality and image generation across both datasets and tasks (conditional and unconditional generation). Notably, this improvement is achieved without introducing any additional learnable parameters.

# 6 DISCUSSION ON FEATURE RANK

In this section, we try to answer a key question: How and why do residual connections impact the abstraction level of the deeper layers in a neural network? We delve deeper into how our design reinforces the low-rank bias of neural networks and try to connect our method to ideas in existing works (Huh et al., 2021). To this end, we visualize the training dynamics of our method and analysis the feature rank of our approach to provide a holistic analysis.

**Low-rank simplicity bias.** Huh et al. (2021) investigate the low-rank simplicity bias in deeper feed-forward neural networks, which drives neural networks to find low-rank solutions. At the same time, they make an empirical observation that deeper residual networks do not show a similar rank contracting behavior.

**Effective rank.** For analysis purpose, Huh et al. (2021) quantify the rank of the learned representation using the *effective rank*, which is a continuous measure. For a matrix $A \in \mathbb{R}^{m \times n}$, the effective rank $\rho(A)$ is defined as the Shannon entropy of the normalized singular values (Roy & Vetterli, 2007):

$$\rho(A) = - \sum_{i}^{\min(n,m)} \bar{\sigma}_i \log \bar{\sigma}_i, \qquad (6)$$

where $\bar{\sigma}_i = \sigma_i / \sum_j \sigma_j$ denotes the $i^{\text{th}}$ normalized singular value. Intuitively, this measure is small when a few singular values dominate and large when singular values are evenly spread, hence giving a good continuous approximation for matrix rank.

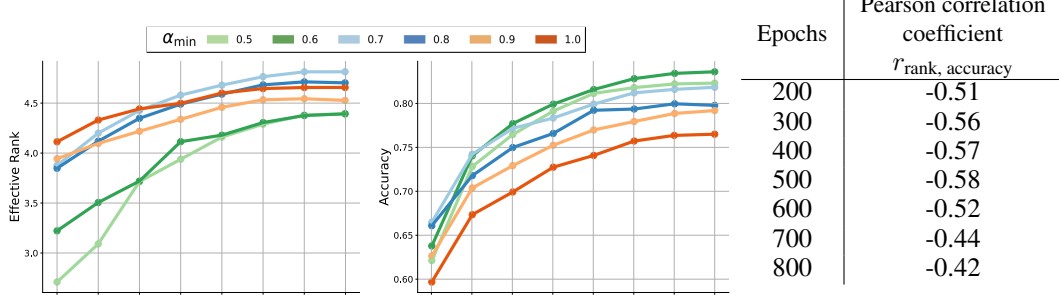

(a) Effective Rank of MAE over time. (b) Linear probing accuracy of MAE.

| Epochs | Pearson correlation coefficient |
|---|---|
| | $r_{\text{rank, accuracy}}$ |
| 200 | -0.51 |
| 300 | -0.56 |
| 400 | -0.57 |
| 500 | -0.58 |
| 600 | -0.52 |
| 700 | -0.44 |
| 800 | -0.42 |

(c) Pearson correlation coefficient between effective rank and probing accuracy, across training epochs.

Figure 5: For MAE pretrained on ImageNet-100, we present visualizations of (a) the training dynamics of the effective rank for different values of $\alpha_{\min}$, (b) the linear probing accuracy for various $\alpha_{\min}$, and (c) the Pearson correlation coefficient between feature rank and probing accuracy, demonstrating that a lower effective feature rank is associated with better performance.

In the following subsections, to compute the effective rank, we use the singular values from the covariance matrix $A_{\theta}$ of the last-layer features, where $A_{\theta}(i, j)$ denotes the covariance of the learned class tokens for the $i^{\text{th}}$ and $j^{\text{th}}$ samples.

Inspired by their analysis, *we conjecture that the improvement to feature learning capability of our method can mainly be attributed to the decayed identity shortcuts promoting low-rank features at the bottleneck.* We measure the training dynamics of the models presented in Table 2 (MAEs trained on ImageNet-100) in terms of accuracy and the effective rank. In Figure 5c, we quantify the correlation between effective feature rank and probing accuracy using Pearson correlation coefficient to validate the hypothesis.

In Figures 5a and 5b, we present the training dynamics of our model, highlighting effective rank and linear probing accuracy for different values of $\alpha_{\min}$. During the early epochs, models with lower $\alpha_{\min}$ tend to exhibit both lower effective rank and higher probing accuracy, supporting our hypothesis. As training progresses, the correlations between $\alpha_{\min}$ and effective rank become less precise. We suspect this is due to the training dynamics, where models with lower $\alpha_{\min}$ experience faster growth in feature rank, reflecting the model's effort to compensate for the decay factors. Despite this, we can still conclude that lower $\alpha_{\min} = [0.5 - 0.6]$ results in lower feature rank and better probing accuracy compared to higher $\alpha_{\min} = [0.7 - 1.0]$.

## 7 CONCLUSION

Huh et al. (2021) raise a key insight in their work – that how a neural network is parameterized matters for fitting the data – and investigate the inductive low-rank bias of stacking more linear layers in a network.

In this work, we observe that the ubiquitous residual network (He et al., 2016) may not be the ideal network parametrization for representation learning and propose a modification of the shortcut path in residual blocks that significantly improves unsupervised representation learning. We explore the connection between our reparameterization of the residual connection and the effective rank of the learned features, finding a correlation between good representations and low-rank representations.

Our work calls into question a fundamental design choice of neural networks that has been used in many modern architectures. By rethinking this choice, the door is open for further reparametrizations and improvements to unsupervised representation learning. The results we show provide a prompt for more extensive investigations into the connection between low effective rank and high-quality abstract representations, as well as the exploration of underlying theoretical mechanisms for this relationship.

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
