## A   TRAINING AND EVALUATION DETAILS

### A.1   MODEL TRAINING.

Our training configurations primarily followed the guidelines established by He et al. (2022). In the ImageNet-1K experiment, our model was trained for 800 epochs, utilizing the AdamW Loshchilov & Hutter (2019) optimizer with a constant weight decay of 5e-2 for a batch size of 1024. We set the maximum learning rate to 6e-4. Initially, the learning rate started at 0 and linearly increased to its maximum over the first 40 epochs, after which it followed a cosine schedule to gradually decrease to zero by the end of the training period. It is worth noting that the learning rate per sample, or effective learning rate, in our setup matched that of He et al. (2022), although our maximum learning rate was set lower due to our batch size being a quarter of theirs. We applied random resizing, cropping, and horizontal flipping during training as part of our augmentation scheme. To enhance the quality of the learned representations in most experiments, we employed the normalized pixel loss, as proposed by He et al. (2022). In the ImageNet-100 experiment, we employed the identical training configuration used in the ImageNet-1K experiments. We train our model with 4 NVIDIA A40 GPUs and a completed trianing usually takes 20 hours on ImageNet-100 and 200 hours on ImageNet-1k.

### A.2   EVALUATION WITH LINEAR PROBING.

For the ImageNet-1k dataset, we use the exact same evaluation protocols employed in He et al. (2022), which includes random data augmentation.

For the ImageNet-100 dataset, we employed a simpler evaluation protocol: We train the linear classifier with a batch size of 1024 for 200 epochs, where the learning rate starts at 1e-2 and then decays towards 0 using a cosine scheduler. During this evaluation, we do not apply any data augmentation.

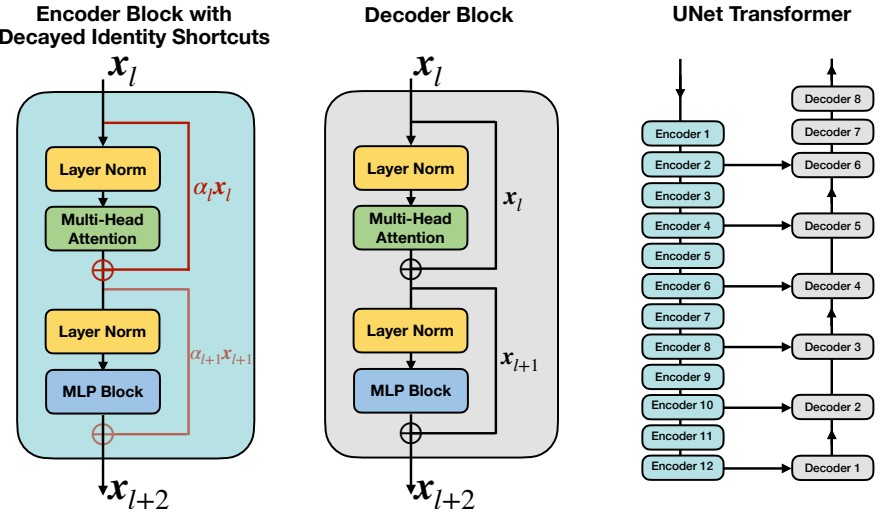

Figure 6: We present our enhanced UNet Transformer architecture for Masked Auto-encoder. (1) **Left**: Our customized encoder blocks, equipped with our proposed decay identity shortcuts. (2) **Middle**: Standard transformer blocks as the decoder blocks. (3) **Right**: We incorporate the decay identity shortcuts exclusively within the encoder blocks of our UNet transformer and employ standard transformer blocks for the decoder. To support abstract representation learning at the bottleneck, *i.e.,* the last layer of the Encoder 12, we adopt the UNet Ronneberger et al. (2015) architecture and create skip connections that transmit every other encoder feature directly to the decoder.

### A.3   MODIFIED ARCHITECTURE

We present a visualization of our UNet transformer design, as outlined in Section 3.2, in Fig. 6. It's important to note that decayed identity shortcuts are exclusively implemented within the encoder

block. Additionally, we establish skip connections from alternating blocks in the encoder to the decoder, following the UNet Ronneberger et al. (2015) architecture's design principles.

# B FURTHER EXPERIMENTS

## B.1 FURTHER ABLATION OF MAXIMUM DECAY RATE ON IMAGENET-1K

| Dataset | Backbone $\alpha_{\min}$ | 0.5 | 0.6 | 0.7 | 0.8 | 0.9 | 1.0 |
|---------|--------------------------|------|------|------|------|------|------|
| ImageNet-1K | ViT-B/16 | 69.8 | **72.7** | 68.9 | - | - | - |
| ImageNet-100 | ViT-B/16 | 82.3 | **83.6** | 81.8 | 79.8 | 79.2 | 76.5 |
| ImageNet-100 | ViT-S/16 | **78.6** | 78.5 | 78.1 | 75.2 | 73.5 | 69.2 |

Table 5: **Linear probing accuracy of our method by varying $\alpha_{\min}$, the architecture size, and the dataset.** We extend the ablation studies detailed in Table. 2 by including linear probing results for the ImageNet-1K dataset. The results indicate that the most favorable $\alpha_{\min}$ consistently falls within the range of $[0.5, 0.6]$ for ImageNet-1K and ImageNet-100 experiments.

We present the results of ablating the choices of $\alpha_{\min}$ on ImageNet-1K dataset in Table.5. From the table, we show that the optimal $\alpha_{\min}$ for ViT-B/16 on ImageNet-100 matches the optimal one for ImageNet-1K, while a lower $\alpha_{\min}$ is preferred for a smaller architecture ViT-S/16.

## B.2 RECONSTRUCTION QUALITY.

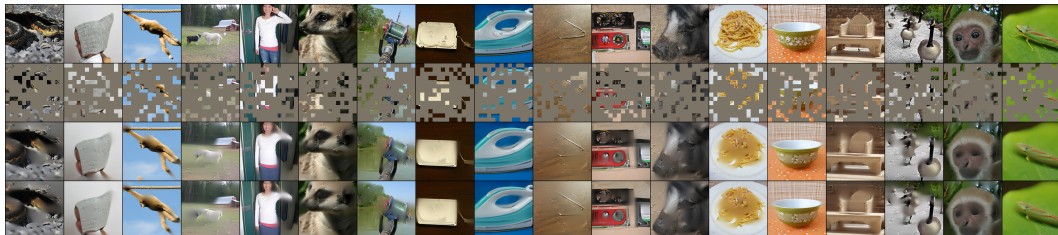

Figure 7: **Qualitative comparison of images reconstructed by MAE with and without our method.** We observe our method learns features with higher linear probing accuracy without compromising reconstruction quality. Row 1: ground truth test image. Row 2: images masked at 75%. Row 3: reconstructions with our method. Row 4: reconstructions with baseline MAE.

We qualitatively evaluate test images reconstructed by an MAE using our framework and images reconstructed by the original MAE. We show the reconstructed images in Figure 7. While the focus of our work is entirely to improve the representations learned by an encoder, we observe that our framework does not harm the reconstructions. Hence, there is no qualitative tradeoff for our increase in linear probing accuracy.

## B.3 ABSTRACTION AND LOW-RANK IN THE SUPERVISED SETTING

In this experiment, we modify the standard ResNet-18 model to experiment with different depth models. By default, the ResNet-18 has a total of 8 residual blocks that are equally distributed into 4 layers. To increase model depth, we repeat residual blocks in the 3rd layer to obtain models varying between 8 and 16 total layers. At convergence, we observe that the models of different depths achieve a similar test accuracy. However, despite similar accuracies, in Figure 8a, which visualizes the effective rank over depth for different values of $\alpha_{\min}$, we see that the effective rank decreases over depth. Furthermore, smaller values of $\alpha_{\min}$ consistently lead to features with lower effective rank.

Next, in Figure 8b, we try to verify our conjecture by visualizing the evolution of effective rank during training when choosing different $\alpha_{\min}$ in our method. For this experiment, we choose to train the standard ResNet-18 using our decayed identity shortcuts. In this setup, we observe that

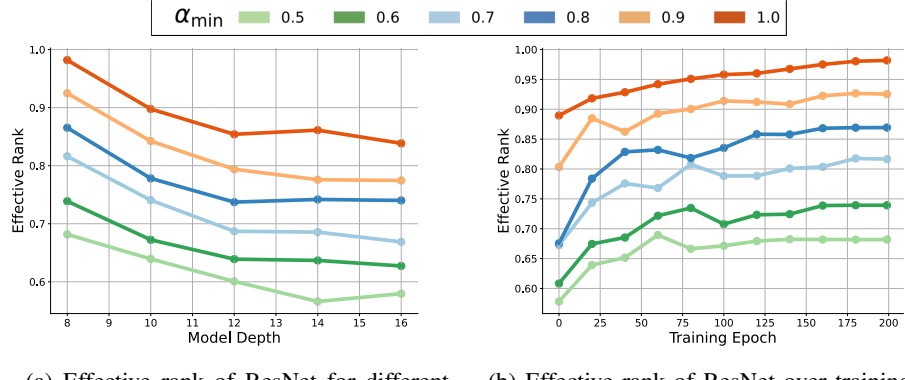

(a) Effective rank of ResNet for different depths at the convergence of the training.

(b) Effective rank of ResNet over training epoch.

Figure 8: **Dynamics of the feature rank in the supervised setup.** We train ResNet models for a supervised classification task on a small subset of ImageNet. And visualize (a) effective rank across different depths at convergence and (b) training dynamics of effective rank over time for various $\alpha_{\min}$. In (a) we see that at convergence, our method consistently decreases the feature rank with various depth and, in (b), this pattern is also shown for standard ResNet model at every stage of training.

the optimal choice of $\alpha_{\min}$ slightly improves the test accuracy of the classification network: 94.4% with $\alpha_{\min} = 0.7$ *vs.* 93.6% with $\alpha_{\min} = 1.0$. We observe that the effective rank of the final features decreases with decreasing $\alpha_{\min}$. This supports our hypothesis that (1) decayed identity shortcuts substantially decrease the rank of bottleneck features and (2) decreasing feature rank may help improve learned features.

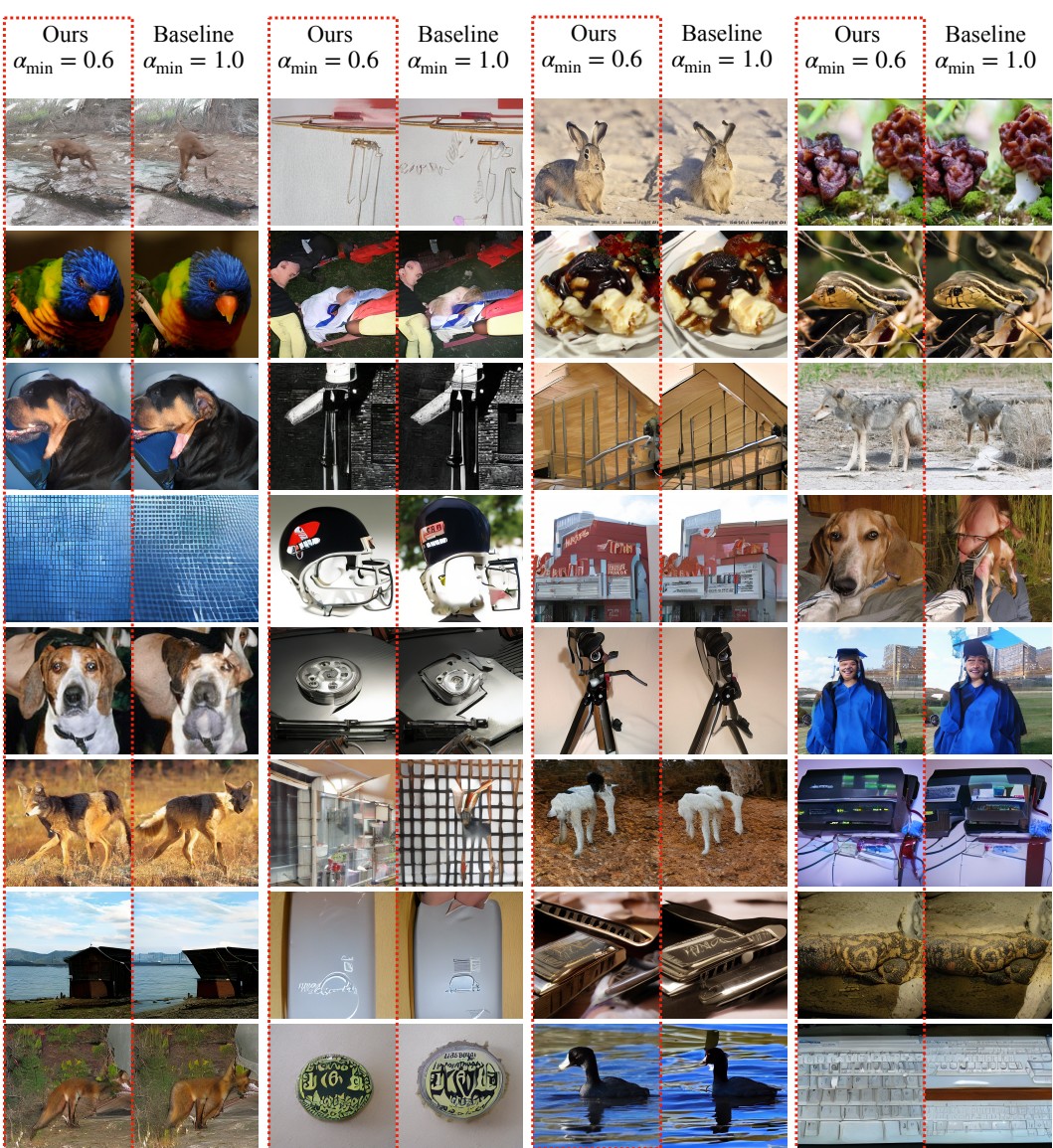

Figure 9: **Qualitative comparison of images generated by diffusion models.** Our method, decayed identity shortcuts with $\alpha_{\min} = 0.6$, shows improved representation learning and produces higher-quality generated images compared to the baseline, which employs full residual connections ($\alpha_{\min} = 1.0$).