# OpenReview forum: "Residual Connections Harm Generative Representation Learning"
_ICLR.cc/2025/Conference — Submitted to ICLR 2025_

### Official Review · Reviewer_saRC · 2024-10-31

**Soundness:** 2
**Presentation:** 3
**Contribution:** 2
**Rating:** 6
**Confidence:** 3

**Summary:**

This paper proposes adjusting the skip connection for generative models by introducing a weight decay. In particular, skip connections at later layers of the network are multiplied by a $\alpha \leq 1$ that decreases monotonically as a function of the layer index. By doing so, the experiments in this paper shows that the learnt representation in generative models improve.

**Strengths:**

The main strengths of this work are:

1- Simplicity of the proposed method: a simple adjustment to the network architecture.

2- Experiments are carried out on different generative models and different settings.

3- The paper is generally well-written and easy to follow.

**Weaknesses:**

Despite the strengths of this work, there are a few concerns that need to be resolved before accepting this work. The order presents the the importance of the concern based on which I rated this paper.

1- Despite the simplicity of the approach, there is no clear methodology on how to choose $\alpha_{\text{min}}$. For example,  in section 4.2, experiments are carried out with $\alpha_{\text{min}}=0.7$, while Table 2 shows that $\alpha_{\text{min}}=0.6$ works the best with ViT-B. This is a key concern as one can choose a suboptimal value of $\alpha_{\text{min}}$ that results in Even a worse performance than just setting $\alpha_{\text{min}}=1$ (i.e. naive residual connection). A discussion on how to properly set $\alpha_{\text{min}}$ should be a core part of the methodology section. Preferably, I suggest the authors to provide systematic analysis of how $\alpha_{\text{min}}$ affects performance across different model sizes and tasks.

2- While the experiments are carried out on both MAE and Diffusion models, the results in this work need to show experiment on  training a generative model with and without the proposed method. For example, a good experiment to show this is to revisit Table and show for each method the impact of the proposed method.

3- The third or of Table 3 (b) raise a critical question: Since weighing both the terms with a simple $\sqrt{0.5}$ results in a significant performance gain, does this mean that the baseline is trained sub-optimally?

4- In section 4.2, the experiments shows a better separation in the learnt feature space when employing the proposed method. It would also make the argument stronger if we can quantify the improvement in that regard (e.g. reporting silhouette score).

**Questions:**

After going through the rebuttal, I commend the authors on the efforts put in addressing my comments. Thus, I raise my score from 3 to 6.

---

> ### Author Response · Authors · 2024-11-29
> **Official Comment by Authors (1/2)**
>
> Thank you for your feedback.
>
> - **W1:** Systematical analysis of the model configurations and selection of optimal $\alpha_{\text{min}}$. (also addressed in the general reply)
>
> We provide extensive analysis across multiple model configurations and provide an empirical criterion for selecting the optimal $\alpha_{\text{min}}$.
>
> Specifically, we evaluate different choices of $\alpha_{\text{min}}$ across two axes: feature dimension and encoder depth, and report the results in Table R1. Linear probing accuracy is measured using features extracted from the encoder's final layer.
>
> In Table R2, we extend the analysis by reporting the highest linear probing accuracy across all layers, along with the index of the best-performing layer (displayed in parentheses). Across various configurations, including ViT-Base and ViT-Large, our method consistently outperforms the baseline that employs a full residual connection ($\alpha_{\text{min}} = 1$).
>
> Our findings reveal that the number of encoder layers $L$, rather than the embedding dimension, primarily determines the optimal $\alpha_{\text{min}}$. We attribute this behavior to the scaling effect of the input data on the encoder's final layer, quantified as: $\alpha_L^{\text{eff}} = \prod_{l=1}^L(1 - \frac{(1 - \alpha_{\text{min}})l}{L})$. Deeper models require larger $\alpha_{\text{min}}$ to maintain a consistent cumulative decay effect represented by $\alpha_L^{\text{eff}}$.
>
> Table R3 presents the values of $\alpha_L^{\text{eff}}$ for the best-performing layer under various configurations.
> From these results, we see that selecting $\alpha_{\text{min}}$ to make $\alpha_L^{\text{eff}} \in [0.001, 0.01)$ consistently yields a better result than the baseline model with a full residual connection. Particularly, along with Table R2, we observe that the improvements are robust when $\alpha_L^{\text{eff}}$ falls in this preferred range. Choosing $\alpha_L^{\text{eff}}$, and using the above formula, we can compute $\alpha_{\text{min}}$ for a particular network.
>
> |Feature Dim.|Encoder Depth  |$\alpha_{\text{min}}$ = 0.6|$\alpha_{\text{min}}$ = 0.7 |$\alpha_{\text{min}}$ = 0.8 |$\alpha_{\text{min}}$ = 0.9 |$\alpha_{\text{min}}$ = 1.0 (Baseline)
> |:---:|:---:|:---:|:---:|:---:|:---:|:---:|
> |384|12|**78.5**|78.1|75.2|73.5|69.2
> |768|12|**83.6**|81.8|79.8|79.2|76.5
> |1024|12|**83.2**|82.5|82.1|79.3|78.0
> |768|18|78.5|**85.0**|84.4|81.8|79.2
> |1024|24|66.7|82.4|**84.5**|84.3|81.4
>
> **Table R1: Linear probing accuracy of the final layer of the encoder.** Across multiple model configurations, we report the probing score when applying our method. For a family of architectures, by properly choosing $\alpha_{\text{min}}$, our method leads to a significantly better result than the baseline with the full residual connection.
>
> |Feature Dim.|Encoder Depth |$\alpha_{\text{min}}$ = 0.6|$\alpha_{\text{min}}$ = 0.7 |$\alpha_{\text{min}}$ = 0.8 |$\alpha_{\text{min}}$ = 0.9 |$\alpha_{\text{min}}$ = 1.0 (Baseline)
> |:---:|:---:|:---:|:---:|:---:|:---:|:---:|
> |384|12|**78.5** (12)|78.1 (12)|75.2 (12)|73.5 (12)|69.2 (12)
> |768|12|**83.6** (12)|81.8 (12)|79.8 (12)|79.2 (12)|76.5 (12)
> |1024|12|**83.2** (12)|82.5 (12)|82.1 (12)|79.3 (12)|78.0 (12)
> |768|18|83.5 (14)|**85.0** (18)|84.4 (18)|81.8 (18)|79.2 (18)
> |1024|24|84.3 (17)|**86.0** (19)|84.5 (24)|84.3 (24)|81.4 (24)
>
> **Table R2: Linear probing accuracy of the best performing layer of the encoder.** Across multiple model configurations, we report the highest linear probing accuracy across all layers, along with the index of the best-performing layer (displayed in parentheses). Comparing to Table R1 shows that the best probing score appears at shallower layers for deeper models.
>
> |Feature Dim.|Encoder Depth |$\alpha_L^{\rm{eff}} \in$ (0, 1e-3) |$\alpha_L^{\rm{eff}} \in$ [1e-3, 1e-2) |$\alpha_L^{\rm{eff}} \in$ [1e-2, 1e-1) |$\alpha_L^{\rm{eff}} \in$ [1e-1, 1] |
> |:---:|:---:|:---:|:---:|:---:|:---:|
> |384|12|---|**78.5**|78.1|75.2|
> |768|12|---|**83.6**|81.8|79.8|
> |1024|12|---|**83.2**|82.5|82.1|
> |768|18|78.5|**85.0**|84.4|81.8|
> |1024|24|82.4|**86.0**|---|84.3|
>
> **Table R3: Best linear probing accuracy for various $\alpha_L^{\text{eff}}$.** Results show that for different model configurations, setting $\alpha_{\text{min}}$ such that $\alpha_{L}^{\text{eff}} \in [1\text{e-3}, 1\text{e-2})$ consistently achieves the best performance.
>
> ---
> - **W2:** Comparison to baseline methods
>
> Our original submission contains extensive comparisons to baseline methods in Tables 2,3,4 and Figure 1. Please note that $\alpha_{\text{min}} = 1$ corresponds to the default residual connections (baseline network architecture) in each setting. The requested comparisons are all present in the original submission, and we see in every setting that our approach outperforms the baseline by using smaller values of $\alpha_{\rm min}$.

---

> > ### Author Response · Authors · 2024-11-29
> > **Official Comment by Authors (2/2)**
> >
> > - **W3:** Discussion of the configuration: $\sqrt{0.5}(x_l + f(x_l))$.
> >
> > The baseline in our paper is a model with default residual connections, i.e., $\alpha_{\min} = 1$, which is the standard setup of MAE recognized by the community [1], augmented with skip connection between encoder-decoder.
> >
> > The format $\sqrt{0.5}(x_l + f(x_l))$ used in previous work [2] is somewhat related to our method, but is fundamentally incorrect. Our ablation study presented in Table 3(b) demonstrates that adding decay to $f(x_l)$ has minimal impact on the baseline, whereas applying decay to $x_l$ --- the technique proposed in our paper --- yields the most significant improvements. This suggests that the correct functional form for decay should not have a multiplicative factor scaling  $f(x_l)$.
> >
> > Earlier work stumbled upon a form that happened to work in a specific case, demonstrating improvements exclusively in image generation quality within diffusion models using convolutional neural networks. In contrast, our paper extensively explores the impact of the decay term on $x_l$ across various models and uncovers its underlying mechanism through rank analysis.
> >
> > Furthermore, the additional ablation experiments on varying $\alpha_{\text{min}}$ with model depth (provided above), show that the decay term should vary with network depth. Specifically, the per-layer decay should be calibrated to an effective total decay over the entire network; holding the latter constant for a class of model architectures requires varying the former. Thus, a fixed constant value for $\alpha_{\text{min}}$, such as $\sqrt{0.5}$, cannot be a general-purpose solution.
> >
> > ---
> > - **W4:** Quantitative evaluation of feature separation.
> >
> > Figure 4 of the original submission already does exactly this evaluation, where ours (10.4 mIOU) is better than the baseline MAE (4.1 mIOU).
> >
> > To calculate the silhouette scores as suggested, we randomly select 10 subsets from the entire 1000 possible categories from ImageNet-1k. Then, we perform clustering on the final CLS token using K-Means clustering with K = 10 and calculate the silhouette score based on the K-Means clustering results. This procedure is repeated 10 times, and we report the mean and standard deviation (in parentheses) for both the silhouette score and the accuracy of K-Means clustering.
> >
> > |Method|Silouette Score $\uparrow$ |K-Means Accuracy $\uparrow$|
> > |--|---|---|
> > |Ours|**0.181** (0.003)	| **70.48** (0.07)|
> > |MAE |0.167 (0.001)|	24.78 (0.02)|
> >
> > **Table R11: Quantitative evaluation of feature separation.**
> >
> > [1] He et al., Masked Autoencoders Are Scalable Vision Learners. CVPR, 2022.
> >
> > [2] Karras et al., Progressive Growing of GANs for Improved Quality, Stability, and Variation. ICLR, 2018.

---

> > > ### Author Response · Authors · 2024-12-01
> > >
> > > Thank you for your feedback! As we approach the deadline for the discussion period, we kindly ask if you have any follow-up questions or concerns that require further clarification. If our responses have addressed your concerns, we would greatly appreciate it if you could consider revising your rating. Thank you!
> > >
> > > To further support the evaluation of our method, we have also assessed feature quality using K-NN on the ImageNet-1K dataset.
> > >
> > > ||MAE|Ours|MoCo-V3|
> > > |:---:|:---:|:---:|:---:|
> > > |Top1 ACC|27.44| 63.92|66.57|
> > > |Top5 ACC|45.33| 83.02|83.09|
> > >
> > > **Table R12.** K-Nearest Neighbor (K=20) accuracy on ImageNet-1K. Our method significantly improves upon the MAE baseline and approaches the performance of MoCo-V3.

---

### Official Review · Reviewer_Kv2k · 2024-11-04

**Soundness:** 2
**Presentation:** 2
**Contribution:** 2
**Rating:** 5
**Confidence:** 4

**Summary:**

The authors proposed decayed identity shortcuts in residual networks, promoting the gradual development of feature abstractions. The author hypothesized that injecting shallower representations into deep layers with residual connections can be harmful by reducing the capacity for abstract learning. Then the authors conducted experiments to show this improves MAE by 4% and also benefits generation with diffusion models.

**Strengths:**

S1. The method is straightforward and easy to follow.

S2. The intuition is reasonable, aligns with some hierarchical representation learning and generation work.

S3. Dive into both representation learning and generative modelling.

**Weaknesses:**

W1. The experiments on representation learning are only on MAEs. In the literature, MAE is not best performing with linear probing and showed to work better in a finetuned manner, more experiments on different models would be greatly beneficial for the paper. No evidence this works universally across models. And the finetuned MAE with decayed identity shortcuts is no better than finetune MAE.

W2. Probing the inner state of different layers could enhance explainability and align with the intuition that underpins the work, providing deeper insights into the model’s internal processing, e.g., using representations from previous layers rather than the final layer, etc.

W3. Introducing an extra hyperparameter that needs to be justified/selected when using.

**Questions:**

Q1. What about a cosine decay schedule? Compared to linear.

Q2. How is this fixed hyperparameter alpha compared to a learnable parameter with regularization and constraints?

Q3. How to compare the idea of the hierarchical representation learning with Matryoshka representation learning? Instead of doing it hierarchically, matryoshka rep learning embeds different virtual concepts into different chunks of embeddings.


[C1] Matryoshka representation learning., Kusupati, Aditya, et al., NeurIPS 2022

---

> ### Author Response · Authors · 2024-11-28
> **Official Comment by Authors (1/3)**
>
> Thank you for your feedback.
>
> - **W1:** The experiments on representation learning are only on MAEs.
>
> **Experiments on Diffusion models.** We demonstrate that our method enhances both representation learning and image generation in diffusion models.
>
> Specifically, Figure 1 and Table 4 of our submission illustrate improvements in representation learning with diffusion models using ViTs and CNNs: (a) For diffusion models with ViT, we increase linear probing accuracy from 72.8\% to 76.1\% and improve FID from 6.93 to 4.98 (conditional generation) and 44.40 to 40.96 (unconditional generation) on ImageNet-100; (b) For diffusion models with CNN, we increase linear probing accuracy from 62.47\% to 66.86\% with FID improving from 14.34 to 8.99.
>
> **Fine-tuning performance.** (also answered in responses to Reviewers uy11 and Hwrn)
>
> Our primary focus is on representation learning. End-to-end fine-tuning primarily evaluates the utility of features as initialization for downstream tasks, which is influenced by various factors such as augmentation strategy and layer-wise learning rates in the fine-tuning stage, rather than directly measuring the quality of the learned representations. In contrast, linear probing offers a more direct assessment of representation quality by utilizing a simple linear projection to adapt the features for downstream applications.
>
> Moreover, the results from end-to-end fine-tuning do not accurately reflect the quality of unsupervised representations, as the fine-tuning process significantly alters the original features. For example, ADDP [1] achieves only 23.8\% accuracy in linear probing, highlighting poor representation learning. However, its accuracy improves significantly to 85.9\% after end-to-end fine-tuning, illustrating the substantial transformation of representations during fine-tuning.
>
> |Model|Ours|MAE|BEiT|ADDP[1]|
> |:---:|:---:|:---:|:----:|:---:|
> |Fine-tune|82.9|83.6|83.2|85.9|
> |Linear Probe|72.7|67.8|56.7|23.8|
>
> **Table R9: Comparison of fine-tuning and linear probing for MAE models on ImageNet-1k.**
>
> ---
>
> - **W2:** Linear probing of the inner state of different layers
>
> Thank you for the suggestions! Here we show the probing performance of $\alpha_{\text{min}} = 0.6$ for ViT-B/16 and $\alpha_{\text{min}} = 0.7$ (the best configuration for each model) for ViT-L/16 for ImageNet-100 experiments.  In the standard model, such as ViT-B, the top probing scores arise from the last layer; In a much deeper model like ViT-L, the optimal features are found in earlier layers (specifically, layer 19).
>
> As depth increases, the optimal features are likely to reside in the earlier layers because the effective decayed weight, calculated as $\prod_{i=1}^l \alpha_i$, decreases exponentially with greater layer depth. In response to the next question, we show with additional experiments (Table R2) that the optimal rate consistently remains within a narrow range.
>
> |Layer Index| 1|2|3|4|5|6|7|8|9|10|11|12|
> |--|--|--|--|--|--|--|--|--|--|--|--|--|
> | ViT-B/16 |24.5|34.8|40.8|42.9|45.5|54.5|62.8|68.9|75.9|80.5|82.6|**83.2**|
>
> |Layer Index| 1|3|5|7|9|11|13|15|17|18|19|21|23|24|
> |--|--|--|--|--|--|--|--|--|--|--|--|--|--|--|
> | ViT-L/16 |26.4|40.5|45.7|52.4|59.1|69.0|75.1|78.0|83.4|85.9|**86.0**|85.6|84.2|82.8|
>
> **Table R10: Linear probing accuracy for inner layers**
>
> ---
>
> - **W3:**  Hyper-parameter selection (also addressed in our general reply)
>
> Our method relies on a single hyperparameter $\alpha_{\text{min}}$, resulting in a minimum modification to the generative representation learning framework, while achieving significant enhancement compared to the baseline and the related work (e.g., I-JEPA).
>
> As shown in Table 2 of our paper, the optimal $\alpha_{\text{min}}$ is about 0.6 for both ViT-B and ViT-S models, highlighting the robustness of our technique. We provide additional results for various model configurations and an empirical guideline for optimal $\alpha_{\text{min}}$ selection. Even when increasing the encoder from 12 to 24 layers and the feature dimension from 768 to 1024, the performance of the optimal layers (see Table R2) is largely unaffected by the choice of $\alpha_{\rm{min}}$.
>
> Specifically, we evaluate different choices of $\alpha_{\text{min}}$ across two axes: feature dimension and encoder depth, and report the results in Table R1. Linear probing accuracy is measured using features extracted from the encoder's final layer.
>
> In Table R2, we extend the analysis by reporting the highest linear probing accuracy across all layers, along with the index of the best-performing layer (displayed in parentheses).
>
> Across various configurations, including ViT-Base and ViT-Large, our method consistently outperforms the baseline that employs a full residual connection ($\alpha_{\text{min}} = 1$).

---

> > ### Author Response · Authors · 2024-11-28
> > **Official Comment by Authors (2/3)**
> >
> > These results reveal that the number of encoder layers $L$, rather than the embedding dimension, primarily determines the optimal $\alpha_{\text{min}}$. We attribute this behavior to the scaling effect of the input data on the encoder's final layer, quantified as: $\alpha_L^{\text{eff}} = \prod_{l=1}^L(1 - \frac{(1 - \alpha_{\text{min}})l}{L})$. Deeper models require larger $\alpha_{\text{min}}$ to maintain a consistent cumulative decay effect represented by $\alpha_L^{\text{eff}}$.
> >
> > Table R3 presents the values of $\alpha_L^{\text{eff}}$ for the best-performing layer under various configurations.
> > From these results, we see that selecting $\alpha_{\text{min}}$ to make $\alpha_L^{\text{eff}} \in [0.001, 0.01)$ consistently yields a better result than the baseline model with a full residual connection. Particularly, along with Table R2, we observe that the improvements are robust when $\alpha_L^{\text{eff}}$ falls in this preferred range. Choosing $\alpha_L^{\text{eff}}$, and using the above formula, we can compute $\alpha_{\text{min}}$ for a particular network.
> >
> > |Feature Dim.|Encoder Depth  |$\alpha_{\text{min}}$ = 0.6|$\alpha_{\text{min}}$ = 0.7 |$\alpha_{\text{min}}$ = 0.8 |$\alpha_{\text{min}}$ = 0.9 |$\alpha_{\text{min}}$ = 1.0 (Baseline)
> > |:---:|:---:|:---:|:---:|:---:|:---:|:---:|
> > |384|12|**78.5**|78.1|75.2|73.5|69.2
> > |768|12|**83.6**|81.8|79.8|79.2|76.5
> > |1024|12|**83.2**|82.5|82.1|79.3|78.0
> > |768|18|78.5|**85.0**|84.4|81.8|79.2
> > |1024|24|66.7|82.4|**84.5**|84.3|81.4
> >
> > **Table R1: Linear probing accuracy of the final layer of the encoder.** Across multiple model configurations, we report the probing score when applying our method. For a family of architectures, by properly choosing $\alpha_{\text{min}}$, our method leads to a significantly better result than the baseline with the full residual connection.
> >
> > |Feature Dim.|Encoder Depth |$\alpha_{\text{min}}$ = 0.6|$\alpha_{\text{min}}$ = 0.7 |$\alpha_{\text{min}}$ = 0.8 |$\alpha_{\text{min}}$ = 0.9 |$\alpha_{\text{min}}$ = 1.0 (Baseline)
> > |:---:|:---:|:---:|:---:|:---:|:---:|:---:|
> > |384|12|**78.5** (12)|78.1 (12)|75.2 (12)|73.5 (12)|69.2 (12)
> > |768|12|**83.6** (12)|81.8 (12)|79.8 (12)|79.2 (12)|76.5 (12)
> > |1024|12|**83.2** (12)|82.5 (12)|82.1 (12)|79.3 (12)|78.0 (12)
> > |768|18|83.5 (14)|**85.0** (18)|84.4 (18)|81.8 (18)|79.2 (18)
> > |1024|24|84.3 (17)|**86.0** (19)|84.5 (24)|84.3 (24)|81.4 (24)
> >
> > **Table R2: Linear probing accuracy of the best-performing layer of the encoder.** Across multiple model configurations, we report the highest linear probing accuracy across all layers, along with the index of the best-performing layer (displayed in parentheses). Comparing to Table R1 shows that the best probing score appears at shallower layers for deeper models.
> >
> > |Feature Dim.|Encoder Depth |$\alpha_L^{\rm{eff}} \in$ (0, 1e-3) |$\alpha_L^{\rm{eff}} \in$ [1e-3, 1e-2) |$\alpha_L^{\rm{eff}} \in$ [1e-2, 1e-1) |$\alpha_L^{\rm{eff}} \in$ [1e-1, 1] |
> > |:---:|:---:|:---:|:---:|:---:|:---:|
> > |384|12|---|**78.5**|78.1|75.2|
> > |768|12|---|**83.6**|81.8|79.8|
> > |1024|12|---|**83.2**|82.5|82.1|
> > |768|18|78.5|**85.0**|84.4|81.8|
> > |1024|24|82.4|**86.0**|---|84.3|
> >
> > **Table R3: Best linear probing accuracy for various $\alpha_L^{\text{eff}}$.** Results show that for different model configurations, setting $\alpha_{\text{min}}$ such that $\alpha_{L}^{\text{eff}} \in [1\text{e-3}, 1\text{e-2})$ consistently achieves the best performance.
> >
> > ---
> > - **Q1:** Experiments with cosine decay schedule (also addressed in our general reply)
> >
> > Thanks for the suggested ablation experiments. We investigate decay scheduler options and perform additional experiments on ImageNet-100, employing a cosine decay scheduler for MAE.
> >
> > |Scheduler|$\alpha_{\rm min}$ = 0.6|$\alpha_{\rm min}$ = 0.7|
> > |---|---|---|
> > Linear |**83.6**|81.8|
> > Cosine |82.8|**82.9**|
> >
> > **Table R4: Ablation on cosine decay scheduler.**
> >
> > The cosine decay schedule is less sensitive to the selection of $\alpha_{\text{min}}$ than the linear decay schedule, yet it achieves lower performance at the optimal $\alpha_{\text{min}}$ for the linear schedule.

---

> > > ### Author Response · Authors · 2024-11-29
> > > **Official Comment by Authors (3/3)**
> > >
> > > - **Q2:** Learnable $\alpha_l$ (also addressed in our general reply)
> > >
> > > We run experiments by making the $\alpha_l$ learnable parameters rather than a fixed constant during training. We add sigmoid activations to constrain the value of $\alpha_l$ between [0,1] and even initialize the learnable parameters with a nearly optimal constant, $\sqrt(0.5)$, based on our prior experiments. No weight decay or penalty is applied for regularizing $\alpha_l$, since we found such regularizations tend to reduce $\alpha_l$ in deeper layers, adversely affecting performance. We show the final $\alpha_l$ for each layer in the following tables:
> > >
> > > |Layer Index| 1|2|3|4|5|6|7|8|9|10|11|12|
> > > |--|--|--|--|--|--|--|--|--|--|--|--|--|
> > > |Attention |0.993|0.947|0.982|0.766|0.992|0.795|0.988|0.849|0.998|0.723|0.811|0.488|
> > > |FFN |0.989|0.926|0.961|0.620|0.961|0.442|0.711|0.322|0.810|0.475|0.637|0.353|
> > >
> > > **Table R5: The learned $\alpha_l$ at convergence of training.**
> > >
> > > |Probing Acc|Linear Schedule $\alpha_{\text{min}}=0.6$|Learnable $\alpha_l$|
> > > |---|---|---|
> > > |ViT-B/16|**83.6**|79.5|
> > >
> > > **Table R6: Comparison of the linear probing accuracy using learnable $\alpha_l$ and our constant linear decay schedule.**
> > >
> > > From the table, we find that, in deeper layers, $\alpha_l$ values decrease and become unstable and the feature quality is even worse than our simpler linear schedule. The unstable nature of learnable $\alpha_l$ might raise the concern for deeper models. This suggests that $\alpha$ is more appropriately regarded as a regularization hyperparameter rather than a learned component of the model.
> > >
> > > ---
> > > - **Q3:** Connection to Matryoshka representation learning
> > >
> > > Thank you for suggesting this work for discussion. Both Matryoshka [2] and our work focus on learning latent representations with specific target structures. However, our work emphasizes designing architectural regularization to learn abstract representations within an unsupervised generative learning framework. In contrast, the Matryoshka paper focuses on learning hierarchical structures using feature dimensions as hierarchical indices in a supervised setting. We will include this discussion in our final version of the paper.
> > >
> > > [1] Tian et al., ADDP: Learning General Representations for Image Recognition and Generation with Alternating Denoising Diffusion Process. ICLR, 2024.
> > >
> > > [2] Kusupati et al., Matryoshka Representation Learning. NeurIPS, 2022.

---

> > ### Comment · Reviewer_Kv2k · 2024-12-01
> >
> > Thanks the authors for the detailed response. I agree that fine-tuning might be affected by some factors. But here's the caveat, it's reasonable to show linear probing results but the choice of base model is questionable and undermines the point, and more experiments on this will benefit the claim more. Also in the foundation model literature, fine-tuning is another perspective to look at. I was hoping to see models other than MAE in this setting.

---

> ### Author Response · Authors · 2024-12-01
>
> We thank you for your comment.
>
> > **it's reasonable to show linear probing results but the choice of base model is questionable and undermines the point**
>
> We respectfully disagree with the statement that ``MAE is a questionable model for showing linear probing results.'' While generative-based frameworks like MAE do not currently achieve state-of-the-art linear probing accuracy compared to contrastive-based methods, they offer a unified framework for both representation learning and data generation. Improving the linear probing performance of generative-based models is a key research focus within the representation learning community. For example, methods such as [1] I-JEPA, [2] Latent-MIM, and [3] SODA target improvements in masked models or diffusion models, while we demonstrate enhancements in both frameworks.
>
> In addition to linear probing, we also evaluate feature quality using K-NN on the ImageNet-1K dataset:
>
> ||MAE|Ours|MoCo-V3|
> |:---:|:---:|:---:|:---:|
> |Top1 ACC $\uparrow$|27.44| 63.92|66.57|
> |Top5 ACC $\uparrow$|45.33| 83.02|83.09|
>
> **Table R12.** K-Nearest Neighbor (K=20) accuracy on ImageNet-1K. Our method significantly improves upon the MAE baseline and approaches the performance of MoCo-V3.
>
> > **I was hoping to see models other than MAE in this setting.**
>
> We agree! In our draft, we already show scalability across (1) MAE, (2) diffusion models (both ViT and CNN architectures), and (3) BEiT (presented in our rebuttal). Across three key metrics—(1) linear probing accuracy, (2) FID for image generation quality, and (3) K-NN accuracy—we demonstrate consistent and significant improvements.
>
> Specifically, compared to the baseline models:
>
> |Method|Model|Dataset|Metric|Baseline|Ours|
> |:---:|:---:|:---:|:---:|:---:|:---:|
> |MAE|ViT-B|ImageNet-1K|KNN(K=20)|27.4|**63.9**|
> |MAE|ViT-B|ImageNet-1K|Linear Probing|67.8|**72.7**|
> |MAE|ViT-L|ImageNet-100|Linear Probing|81.4|**86.0**|
> |MAE|ViT-B|ImageNet-100|Linear Probing|76.5|**83.6**|
> |MAE|ViT-S|ImageNet-100|Linear Probing|69.2|**78.5**|
> |Diffusion|ViT-B|ImageNet-100|Linear Probing|72.8|**76.1**|
> |Diffusion|ViT-B|ImageNet-100|FID|6.93|**4.98**|
> |Diffusion|CNN|CIFAR-100|Linear Probing|62.5|**66.9**|
> |Diffusion|CNN|CIFAR-100|FID|14.34|**8.99**|
> |BEiT|ViT-B|ImageNet-100|Linear Probing|71.8|**74.2**|
>
>
> Although our fine-tuning experiments have not yet shown improvements (with a slight 0.7\% decrease, likely due to suboptimal hyperparameter tuning and augmentation strategies at the fine-tuning stage), our results demonstrate significant advancements across three metrics and multiple frameworks. This highlights the scalability, robustness, and effectiveness of our method. Therefore, we would greatly appreciate it if you could evaluate our contributions more comprehensively and reconsider your rating. We are happy to address any additional concerns. Thank you!
>
>
> [1] Self-Supervised Learning from Images with a Joint-Embedding Predictive Architecture, Mahmoud Assran et al., 2023
>
> [2] Towards Latent Masked Image Modeling for Self-Supervised Visual Representation Learning, Yibing Wei, Abhinav Gupta, Pedro Morgado, 2024.
>
> [3] SODA: Bottleneck Diffusion Models for Representation Learning, Drew A. Hudson et al., 2023

---

### Official Review · Reviewer_Hwrn · 2024-11-04

**Soundness:** 3
**Presentation:** 2
**Contribution:** 2
**Rating:** 5
**Confidence:** 4

**Summary:**

The paper shows that linearly decaying (over layers) the residual term in ViT-like models improves the training of generative models. With a few details, e.g., adding U-Net like skip connections between encoder and decoder, and initializing the final encoder output layer to be zero, the proposed method could successfully stabilize the training despite of the gradual removal of residual connections. Experiments are mainly done in two scenarios: training (a) Masked Autoencoders on ImageNet-1K, and (b) Diffusion models on CIFAR-100 and ImageNet-100. The paper reports that the proposed method improves the linear probing accuracy of both MAE and Diffusion training, as well as FID for Diffusion models at certain choice of decaying hyperparameter.

**Strengths:**

- The paper is well-motivated and easy-to-follow.
- The proposed method is notably simple and easy-to-adapt, which can be a strength in large-scale training regimes where recent generative models are indeed at.
- Rethinking the design space of neural architecture, e.g., ViTs, for generative modeling is a timely research direction.
- Figure 4 effectively highlight the quality of the learned representations with quantitative results.

**Weaknesses:**

- It is still questionable to me whether the proposed method is scalable for deeper networks; in my experience weakening the residual terms does introduce instability of overall training, whereas the paper did not much discuss about when the method fails. For example, Table 2 shows that the optimal hyperparameter of the method varies depending on the depth of the backbone networks, i.e., ViT-S vs. ViT-B. Also, Table 4 seems to show that the method is sensitive to hyperparameter choice depending on datasets. The paper may extend such analysis for wider depths, data, etc., possibly suggesting any guideline in using the method in practice.
- Table 1: The proposed method does not improve MAE on fine-tuning setup, while the performance of MAE was originally focused on after the fine-tuning. Although I agree with the paper’s comment that the fine-tuning performance is harder to compare with, I feel the current results in Table 1 does not fully confirm the main claim, i.e., residual connection harms generative representation learning. The paper may strengthen this point by showing that the method is effective for other generative modeling approaches, e.g., I-JEPA, BEiT, MaskGiT, etc., if applicable.
- The effectiveness of method seems to be dependent to an architectural modification of ViT, i.e., only after adding encoder-decoder skip connections as like U-Net (Table 3a). But such a skip connection may already improve MAE, although I don’t think this point is clearly discussed in the current manuscript.
- (minor) Section 4.2: “Embedding analysis. → Embedding analysis”

**Questions:**

- One alternative of decaying the residual term can be stochastic depth; i.e., by randomly dropping the identity term during training. I am curious that the authors have tested its performance, given that stochastic depth is also well-known to improve generalization of residual network training.
- Table 3b: The comparison shows that an ablation $\sqrt{0.5}(x_l + f(x_l))$ could achieve a quite competitive result to the proposed method. If so, given that the form has been already used in previous works and that it is simpler (e.g., the form does not have hyperparameter), what would be the key benefits of the proposed method over that ablated form?
- Can this method also improve models other than ViTs?

---

> ### Author Response · Authors · 2024-11-28
> **Official Comment by Authors (1/3)**
>
> Thank you for your feedback.
>
> - **W1:** Scalable for deeper networks and the optimal choices of $\alpha_{\text{min}}$.
>
> Our approach effectively scales with both increased model depth and channel size. The decay scheme we propose gradually adjusts the decay factor as depth increases, ensuring a smooth transition in even the deeper models.
>
> To show this, we provide additional results (also in the general reply) across multiple model configurations and provide an empirical criterion for selecting the optimal $\alpha_{\text{min}}$. Specifically, we evaluate different choices of $\alpha_{\text{min}}$ across two axes: feature dimension and encoder depth, and report the results in Table R1. Linear probing accuracy is measured using features extracted from the encoder's final layer.
>
> In Table R2, we extend the analysis by reporting the highest linear probing accuracy across all layers, along with the index of the best-performing layer (displayed in parentheses). Across various configurations, including ViT-Base and ViT-Large, our method consistently outperforms the baseline that employs a full residual connection ($\alpha_{\text{min}} = 1$).
>
> Our findings reveal that the number of encoder layers $L$, rather than the embedding dimension, primarily determines the optimal $\alpha_{\text{min}}$. We attribute this behavior to the scaling effect of the input data on the encoder's final layer, quantified as: $\alpha_L^{\text{eff}} = \prod_{l=1}^L(1 - \frac{(1 - \alpha_{\text{min}})l}{L})$. Deeper models require larger $\alpha_{\text{min}}$ to maintain a consistent cumulative decay effect represented by $\alpha_L^{\text{eff}}$.
>
> Table R3 presents the values of $\alpha_L^{\text{eff}}$ for the best-performing layer under various configurations.
> From these results, we see that selecting $\alpha_{\text{min}}$ to make $\alpha_L^{\text{eff}} \in [0.001, 0.01)$ consistently yields a better result than the baseline model with a full residual connection. Particularly, along with Table R2, we observe that the improvements are robust when $\alpha_L^{\text{eff}}$ falls in this preferred range. Choosing $\alpha_L^{\text{eff}}$, and using the above formula, we can compute $\alpha_{\text{min}}$ for a particular network.
>
> |Feature Dim.|Encoder Depth  |$\alpha_{\text{min}}$ = 0.6|$\alpha_{\text{min}}$ = 0.7 |$\alpha_{\text{min}}$ = 0.8 |$\alpha_{\text{min}}$ = 0.9 |$\alpha_{\text{min}}$ = 1.0 (Baseline)
> |:---:|:---:|:---:|:---:|:---:|:---:|:---:|
> |384|12|**78.5**|78.1|75.2|73.5|69.2|
> |768|12|**83.6**|81.8|79.8|79.2|76.5|
> |1024|12|**83.2**|82.5|82.1|79.3|78.0|
> |768|18|78.5|**85.0**|84.4|81.8|79.2|
> |1024|24|66.7|82.4|**84.5**|84.3|81.4|
>
> **Table R1: Linear probing accuracy of the final layer of the encoder.** Across multiple model configurations, we report the probing score when applying our method. For a family of architectures, by properly choosing $\alpha_{\text{min}}$, our method leads to a significantly better result than the baseline with the full residual connection.
>
> |Feature Dim.|Encoder Depth |$\alpha_{\text{min}}$ = 0.6|$\alpha_{\text{min}}$ = 0.7 |$\alpha_{\text{min}}$ = 0.8 |$\alpha_{\text{min}}$ = 0.9 |$\alpha_{\text{min}}$ = 1.0 (Baseline)
> |:---:|:---:|:---:|:---:|:---:|:---:|:---:|
> |384|12|**78.5** (12)|78.1 (12)|75.2 (12)|73.5 (12)|69.2 (12)
> |768|12|**83.6** (12)|81.8 (12)|79.8 (12)|79.2 (12)|76.5 (12)
> |1024|12|**83.2** (12)|82.5 (12)|82.1 (12)|79.3 (12)|78.0 (12)
> |768|18|83.5 (14)|**85.0** (18)|84.4 (18)|81.8 (18)|79.2 (18)
> |1024|24|84.3 (17)|**86.0** (19)|84.5 (24)|84.3 (24)|81.4 (24)
>
> **Table R2: Linear probing accuracy of the best-performing layer of the encoder.** Across multiple model configurations, we report the highest linear probing accuracy across all layers, along with the index of the best-performing layer (displayed in parentheses). Comparing to Table R1 shows that the best probing score appears at shallower layers for deeper models.
>
> |Feature Dim.|Encoder Depth |$\alpha_L^{\rm{eff}} \in$ (0, 1e-3) |$\alpha_L^{\rm{eff}} \in$ [1e-3, 1e-2) |$\alpha_L^{\rm{eff}} \in$ [1e-2, 1e-1) |$\alpha_L^{\rm{eff}} \in$ [1e-1, 1] |
> |:---:|:---:|:---:|:---:|:---:|:---:|
> |384|12|---|**78.5**|78.1|75.2|
> |768|12|---|**83.6**|81.8|79.8|
> |1024|12|---|**83.2**|82.5|82.1|
> |768|18|78.5|**85.0**|84.4|81.8|
> |1024|24|82.4|**86.0**|---|84.3|
>
> **Table R3: Best linear probing accuracy for various $\alpha_L^{\text{eff}}$.** Results show that for different model configurations, setting $\alpha_{\text{min}}$ such that $\alpha_{L}^{\text{eff}} \in [1\text{e-3}, 1\text{e-2})$ consistently achieves the best performance.

---

> ### Author Response · Authors · 2024-11-28
> **Official Comment by Authors (2/3)**
>
> - **W2:** fine-tuning experiments (also answered in responses to Reviewers uy11 and Kv2k)
>
> Our primary focus is on representation learning. End-to-end fine-tuning primarily evaluates the utility of features as initialization for downstream tasks, which is influenced by various factors such as augmentation strategy and layer-wise learning rates in the fine-tuning stage, rather than directly measuring the quality of the learned representations. In contrast, linear probing offers a more direct assessment of representation quality by utilizing a simple linear projection to adapt the features for downstream applications.
>
> Moreover, the results from end-to-end fine-tuning do not accurately reflect the quality of unsupervised representations, as the fine-tuning process significantly alters the original features. For example, ADDP [1] achieves only 23.8\% accuracy in linear probing, highlighting poor representation learning. However, its accuracy improves significantly to 85.9\% after end-to-end fine-tuning, illustrating the substantial transformation of representations during fine-tuning.
>
> |Model|Ours|MAE|BEiT|ADDP[1]|
> |:---:|:---:|:---:|:----:|:---:|
> |Fine-tune|82.9|83.6|83.2|85.9|
> |Linear Probe|72.7|67.8|56.7|23.8|
>
> **Table R9: Comparison of fine-tuning and linear probing for MAE models on ImageNet-1k.**
>
> - **W2:** experiments with other generative models
>
> We appreciate your recommendation to extend our methods to other generative models. We investigate this by presenting our results for BEiT. Unfortunately, MaskGiT's official training script is not publicly available, and I-JEPA's training objective could lead to collapsed solutions. Their technique of using exponential moving average (EMA) to prevent a trivial solution is inadequate for our proposed low-rank bias. To address this, additional regularization is necessary. Our method is applied to BEiT by adhering to the original setup, incorporating skip connections between the encoder and decoder, and modifying the encoder's residual connection decay as per our approach. We run experiments on ImageNet-100 and report the highest linear probing accuracy across all layers, demonstrating enhanced performance over baseline.
>
> |$\alpha_{\rm min}$ = 0.8|$\alpha_{\rm min}$ = 1.0(Baseline)|
> |:---:|:---:|
> **74.2**|71.8
>
> **Table R8: Results with BEiT.** Our method improves the probing accuracy compared to the baseline with the full residual connection.
>
> ---
> - **W3:** Discussion of encoder-decoder skip connections
>
> We discuss the UNet setup in Lines 256-262 and show the comparison for our method w/ and w/o the UNet setup in Table 3(a). For a fair comparison, competing methods (including the baseline method, equivalent to $\alpha_{\text{min}} = 1.0$) also have encoder-decoder skip connections and Table 3(b) shows our method is substantially better than the baseline.
>
> ---
> - **W4:** Typo
>
> Thank you for pointing it out; we will correct it in the final version.
>
> ---
> - **Q1:** Experiments with Stochastic Depth (also covered in the general reply)
>
> Training with stochastic depth achieves the opposite of our desired effect. Stochastic depth randomly drops a parameterized layer, keeping the residual connection that runs parallel to it. This increases the relative contribution of the residual link to downstream layer inputs, thus amplifying the importance of that residual connection. We want to decay, not amplify, residual connection strength.
>
> To verify that dropping layers, as in stochastic depth, is not a replacement for decaying identity shortcuts, we perform the following experiment. We keep the same training configurations for our ImageNet-100 MAE experiments and replace our proposed decayed identity shortcuts with randomly dropping network blocks as implemented in [2]. Following [2], we use the same linear decay scheduler, and layer $l$ is dropped with a chance of $\frac{l}{L}\cdot \text{Drop}_{\text{min}}$ during training. We report the linear probing of final representations as follows:
>
> |$\text{Drop}_{\text{min}}$|0.0(Baselines)|0.1|0.2|0.3|0.4|
> |:---:|:---:|:---:|:---:|:---:|:---:|
> |Probing Acc|**76.5**|76.4|76.3|74.7|73.5
>
> **Table R7: Comparison to stochastic depth.**
>
> The more we increase the drop chance, the more stochastic depth harms generative representation learning.

---

> > ### Author Response · Authors · 2024-11-28
> > **Official Comment by Authors (3/3)**
> >
> > - **Q2:** Discussion of the configuration: $\sqrt{0.5}(x_l + f(x_l))$
> >
> > The format $\sqrt{0.5}(x_l + f(x_l))$ used in previous work is somewhat related to our method, but is fundamentally incorrect. Our ablation study presented in Table 3(b) demonstrates that adding decay to $f(x_l)$ has minimal impact on the baseline, whereas applying decay to $x_l$ --- the technique proposed in our paper --- yields the most significant improvements. This suggests that the correct functional form for decay should not have a multiplicative factor scaling  $f(x_l)$.
> >
> > Earlier work stumbled upon a form that happened to work in a specific case, demonstrating improvements exclusively in image generation quality within diffusion models using convolutional neural networks. In contrast, our paper extensively explores the impact of the decay term on $x_l$ across various models and uncovers its underlying mechanism through rank analysis.
> >
> > Furthermore, the additional ablation experiments on varying $\alpha_{\text{min}}$ with model depth (provided above), show that the decay term should vary with network depth. Specifically, the per-layer decay should be calibrated to an effective total decay over the entire network; holding the latter constant for a class of model architectures requires varying the former. Thus, a fixed constant value for $\alpha_{\text{min}}$, such as $\sqrt{0.5}$, cannot be a general-purpose solution.
> >
> > ---
> > - **Q3:** Extension to non-ViT models
> >
> > Yes, our method works for non-ViT models! In Table 4 of our submission, we demonstrate improvements in convolutional neural networks for diffusion models on CIFAR-100, showing enhancements over the baseline in both representation learning and image generation quality. Additionally, in the supplementary material (Appendix B.3), we provide a rank analysis for supervised training with convolutional neural networks, highlighting the reduced feature rank achieved with our method.
> >
> > [1] Tian et al., ADDP: Learning General Representations for Image Recognition and Generation with Alternating Denoising Diffusion Process. ICLR, 2024.
> >
> > [2] Huang et al., Deep Networks with Stochastic Depth. ECCV, 2016.

---

> > > ### Author Response · Authors · 2024-12-01
> > >
> > > Thank you for your feedback! As we approach the deadline for the discussion period, we kindly ask if you have any follow-up questions or concerns that require further clarification. If our responses have addressed your concerns, we would greatly appreciate it if you could consider revising your rating. Thank you!
> > >
> > > To further support the evaluation of our method, we have also assessed feature quality using K-NN on the ImageNet-1K dataset.
> > >
> > > ||MAE|Ours|MoCo-V3|
> > > |:---:|:---:|:---:|:---:|
> > > |Top1 ACC|27.44| 63.92|66.57|
> > > |Top5 ACC|45.33| 83.02|83.09|
> > >
> > > **Table R12.** K-Nearest Neighbor (K=20) accuracy on ImageNet-1K. Our method significantly improves upon the MAE baseline and approaches the performance of MoCo-V3.

---

### Official Review · Reviewer_uy11 · 2024-11-04

**Soundness:** 3
**Presentation:** 3
**Contribution:** 3
**Rating:** 6
**Confidence:** 4

**Summary:**

The paper proposes two architectural changes to ViT that can greatly improve MAE's linear probing performance: (1) decreasing shortcut branch weight in deeper layers and (2) skip connection from encoder layers to decoder layers. With both changes, the authors observe better representations, evaluated with qualitative visualizations and linear probing accuracy. The authors showed that similar modifications can also improve diffusion model performance, and used effective rank as a lens to experimentally analyze & intuitively explain the performance gain.

----

I have read the authors' response and decide to keep my original evaluation.

**Strengths:**

+ Simple modification with effective performance gains.
+ Clear writing of the method & motivation.
+ Diverse experiments for MAE, diffusion models, and effective rank analysis.

**Weaknesses:**

+ The usual way of using MAE is not linear probing, but fine-tuning (or maybe nonlinear probing). The proposed method, unfortunately, reduces fine-tuning performance (Table 1).

**Questions:**

1. The proposed method reduces fine-tuning performance of MAE (which is its strongest suit) and doesn't improve linear probing to be on-par with SOTA (e.g., DINOv2). Given this, have the authors explored applying the changes to DINO? If it yields benefits, it could significantly improve the paper's value.

2. If the residual connection fundamentally limits effective rank, and thus hurts representation quality, why do objectives like DINO not have such issues?

---

> ### Author Response · Authors · 2024-11-28
>
> Thank you for your feedback.
>
> - **W1:** Fine-tuning performance (also answered in responses to Reviewers Hwrn and Kv2k)
>
> Our primary focus is on representation learning. End-to-end fine-tuning primarily evaluates the utility of features as initialization for downstream tasks, which is influenced by various factors such as augmentation strategy and layer-wise learning rates in the fine-tuning stage, rather than directly measuring the quality of the learned representations. In contrast, linear probing offers a more direct assessment of representation quality by utilizing a simple linear projection to adapt the features for downstream applications.
>
> Moreover, the results from end-to-end fine-tuning do not accurately reflect the quality of unsupervised representations, as the fine-tuning process significantly alters the original features. For example, ADDP [1] achieves only 23.8\% accuracy in linear probing, highlighting poor representation learning. However, its accuracy improves significantly to 85.9\% after end-to-end fine-tuning, illustrating the substantial transformation of representations during fine-tuning.
>
> |Model|Ours|MAE|BEiT|ADDP[1]|
> |:---:|:---:|:---:|:----:|:---:|
> |Fine-tune|82.9|83.6|83.2|85.9|
> |Linear Probe|72.7|67.8|56.7|23.8|
> **Table R9: Comparison of fine-tuning and linear probing for MAE models on ImageNet-1k.**
>
> ---
> - **Q1-Q2:** Applying our method to DINO.
>
> Different representation learning approaches exhibit varying preferences for feature rank. Generative representation learning methods, such as MAE and Diffusion, benefit from low-rank regularization. In contrast, contrastive and discriminative methods, including DINO, favor high-rank features due to mechanisms like the repulsive term in the denominator or excessive cluster centers like DINO. RankMe [2] supports this hypothesis, demonstrating that the performance of contrastive models improves as they learn higher-rank features.
>
> Our approach, which induces low-rank features, is therefore not well-suited for contrastive learning and could negatively impact its performance. As highlighted in our title, our method is specifically designed for generative representation learning, which imposes fewer assumptions on data distribution compared to contrastive learning, and is not intended for the contrastive learning framework.
>
> [1] Tian et al., ADDP: Learning General Representations for Image Recognition and Generation with Alternating Denoising Diffusion Process. ICLR, 2024.
>
> [2] Garrido et al., RankMe: Assessing the Downstream Performance of Pretrained Self-Supervised Representations by Their Rank. ICML, 2023.

---

> > ### Author Response · Authors · 2024-12-01
> >
> > Thank you for your feedback! As we approach the deadline for the discussion period, we kindly ask if you have any follow-up questions or concerns that require further clarification. If our responses have addressed your concerns, we would greatly appreciate it if you could consider revising your rating. Thank you!
> >
> > To further support the evaluation of our method, we have also assessed feature quality using K-NN on the ImageNet-1K dataset.
> >
> > ||MAE|Ours|MoCo-V3|
> > |:---:|:---:|:---:|:---:|
> > |Top1 ACC|27.44| 63.92|66.57|
> > |Top5 ACC|45.33| 83.02|83.09|
> >
> > **Table R12.** K-Nearest Neighbor (K=20) accuracy on ImageNet-1K. Our method significantly improves upon the MAE baseline and approaches the performance of MoCo-V3.

---

### Author Response · Authors · 2024-11-28
**General Reply (1/3)**

We thank the reviewers for their feedback. We appreciate the recognition of the **conciseness** (uy11, Hwrn, Kv2k, saRC) and **effectiveness** (uy11, HWrn, Kv2k) of our proposed method, along with the acknowledgment of our **clear presentation and motivation** (uy11, Hwrn, Kv2k) and **diverse experiments and analysis** (uy11, saRC). In response to reviewer questions, we provide both new experimental results and clarifications.

**New Results:** To demonstrate the scalability and robustness of our method, we present experiments over a broader range of model configurations. We show that the optimal value of our hyper-parameter $\alpha_{\text{min}}$ changes in a consistent manner with network depth: the effective decay rate (cumulative decay over the entire network) is key; once this is found for a model, we can use an empirical formula to select $\alpha_{\text{min}}$ for other model architectures in the same class.

Additionally, we provide ablation studies on different decay schedules, the use of learnable $\alpha_l$, and a comparison to stochastic depth. Our findings indicate that our method, combined with a simpler linear schedule, performs the best. We validate our method on BEiT and show improvement over the baseline. We also include visualizations of our generated images in the updated supplementary material (Appendix Figure 9), making it apparent that our method also improves generation quality.

**Clarifications:** In responses to individual reviewers, we address multiple misconceptions, including overlooked primary experiments on diffusion models and convolutional neural networks, discussions regarding encoder-decoder skip connections, and the baseline models used for comparison.

The results of our new experiments are as follows:

---

> ### Author Response · Authors · 2024-11-28
> **General Reply (2/3)**
>
> ## 1. Effect of $\alpha_{\text{min}}$ for Wider Model Configurations and Automatic Selection of Optimal $\alpha_{\text{min}}$
>
> To assess the compatibility of our proposed method across a broader range of model configurations, we evaluate different choices of $\alpha_{\text{min}}$ across two axes: feature dimension and encoder depth, and report the results in Table R1. Linear probing accuracy is measured using features extracted from the encoder's final layer.
>
> In Table R2, we extend the analysis by reporting the highest linear probing accuracy across all layers, along with the index of the best-performing layer (displayed in parentheses). Across various configurations, including ViT-Base and ViT-Large, our method consistently outperforms the baseline that employs a full residual connection ($\alpha_{\text{min}} = 1$).
>
> These results reveal that the number of encoder layers $L$, rather than the embedding dimension, primarily determines the optimal $\alpha_{\text{min}}$. We attribute this behavior to the scaling effect of the input data on the encoder's final layer, quantified as: $\alpha_L^{\text{eff}} = \prod_{l=1}^L(1 - \frac{(1 - \alpha_{\text{min}})l}{L})$. Deeper models require larger $\alpha_{\text{min}}$ to maintain a consistent cumulative decay effect represented by $\alpha_L^{\text{eff}}$.
>
> Table R3 presents the values of $\alpha_L^{\text{eff}}$ for the best-performing layer under various configurations.
> From these results, we see that selecting $\alpha_{\text{min}}$ to make $\alpha_L^{\text{eff}} \in [0.001, 0.01)$ consistently yields a better result than the baseline model with a full residual connection. Particularly, along with Table R2, we observe that the improvements are robust when $\alpha_L^{\text{eff}}$ falls in this preferred range. Choosing $\alpha_L^{\text{eff}}$, and using the above formula, we can compute $\alpha_{\text{min}}$ for a particular network.
>
> |Feature Dim.|Encoder Depth  |$\alpha_{\text{min}}$ = 0.6|$\alpha_{\text{min}}$ = 0.7 |$\alpha_{\text{min}}$ = 0.8 |$\alpha_{\text{min}}$ = 0.9 |$\alpha_{\text{min}}$ = 1.0 (Baseline)
> |:---:|:---:|:---:|:---:|:---:|:---:|:---:|
> |384|12|**78.5**|78.1|75.2|73.5|69.2
> |768|12|**83.6**|81.8|79.8|79.2|76.5
> |1024|12|**83.2**|82.5|82.1|79.3|78.0
> |768|18|78.5|**85.0**|84.4|81.8|79.2
> |1024|24|66.7|82.4|**84.5**|84.3|81.4
>
> **Table R1: Linear probing accuracy of the final layer of the encoder.** Across multiple model configurations, we report the probing score when applying our method. For a family of architectures, by properly choosing $\alpha_{\text{min}}$, our method leads to a significantly better result than the baseline with the full residual connection.
>
> |Feature Dim.|Encoder Depth |$\alpha_{\text{min}}$ = 0.6|$\alpha_{\text{min}}$ = 0.7 |$\alpha_{\text{min}}$ = 0.8 |$\alpha_{\text{min}}$ = 0.9 |$\alpha_{\text{min}}$ = 1.0 (Baseline)
> |:---:|:---:|:---:|:---:|:---:|:---:|:---:|
> |384|12|**78.5** (12)|78.1 (12)|75.2 (12)|73.5 (12)|69.2 (12)
> |768|12|**83.6** (12)|81.8 (12)|79.8 (12)|79.2 (12)|76.5 (12)
> |1024|12|**83.2** (12)|82.5 (12)|82.1 (12)|79.3 (12)|78.0 (12)
> |768|18|83.5 (14)|**85.0** (18)|84.4 (18)|81.8 (18)|79.2 (18)
> |1024|24|84.3 (17)|**86.0** (19)|84.5 (24)|84.3 (24)|81.4 (24)
>
> **Table R2: Linear probing accuracy of the best-performing layer of the encoder.** Across multiple model configurations, we report the highest linear probing accuracy across all layers, along with the index of the best-performing layer (displayed in parentheses). Comparing to Table R1 shows that the best probing score appears at shallower layers for deeper models.
>
> |Feature Dim.|Encoder Depth |$\alpha_L^{\rm{eff}} \in$ (0, 1e-3) |$\alpha_L^{\rm{eff}} \in$ [1e-3, 1e-2) |$\alpha_L^{\rm{eff}} \in$ [1e-2, 1e-1) |$\alpha_L^{\rm{eff}} \in$ [1e-1, 1] |
> |:---:|:---:|:---:|:---:|:---:|:---:|
> |384|12|---|**78.5**|78.1|75.2|
> |768|12|---|**83.6**|81.8|79.8|
> |1024|12|---|**83.2**|82.5|82.1|
> |768|18|78.5|**85.0**|84.4|81.8|
> |1024|24|82.4|**86.0**|---|84.3|
>
> **Table R3: Best linear probing accuracy for various $\alpha_L^{\text{eff}}$.** Results show that for different model configurations, setting $\alpha_{\text{min}}$ such that $\alpha_{L}^{\text{eff}} \in [1\text{e-3}, 1\text{e-2})$ consistently achieves the best performance.

---

> ### Author Response · Authors · 2024-11-28
> **General Reply (3/3)**
>
> ## 2. Cosine Decay Schedule v.s. Linear Decay Schedule
> We investigate decay schedule options and perform additional experiments on ImageNet-100, employing a cosine decay schedule for MAE.
>
> |Schedule|$\alpha_{\text{min}}$ = 0.6|$\alpha_{\text{min}}$ = 0.7|
> |---|---|---|
> Linear |**83.6**|81.8|
> Cosine |82.8|**82.9**|
>
> **Table R4: Ablation on cosine decay schedule.**
>
> The cosine decay schedule is less sensitive to the selection of $\alpha_{\text{min}}$ than the linear decay schedule, yet it achieves lower performance at the optimal $\alpha_{\text{min}}$ than for the linear schedule.
>
> ## 3. Learnable $\alpha_l$
>
> We run experiments by making the $\alpha_l$ learnable parameters rather than a fixed constant during training. We run experiments with ViT-B/16 on ImageNet-100 dataset and we use the same training configurations for our ImageNet-100 experiment. We add sigmoid activations to constrain the value of $\alpha_l$ between [0,1] and even initialize the learnable parameters with a nearly optimal constant, $\sqrt{0.5}$, based on our prior experiments. No weight decay or penalty is applied for regularizing $\alpha_l$, since we found such regularizations tend to reduce $\alpha_l$ in deeper layers, adversely affecting performance. We show the final $\alpha_l$ for each layer in the following tables:
>
> |Layer Index| 1|2|3|4|5|6|7|8|9|10|11|12|
> |--|--|--|--|--|--|--|--|--|--|--|--|--|
> |Attention |0.993|0.947|0.982|0.766|0.992|0.795|0.988|0.849|0.998|0.723|0.811|0.488|
> |FFN |0.989|0.926|0.961|0.620|0.961|0.442|0.711|0.322|0.810|0.475|0.637|0.353|
>
> **Table R5: The learned $\alpha_l$ at convergence of training.**
>
> |Probing Acc|Linear Schedule $\alpha_{\text{min}}=0.6$|Learnable $\alpha_l$|
> |---|---|---|
> |ViT-B/16|**83.6**|79.5|
>
> **Table R6: Comparison of the linear probing accuracy using learnable $\alpha_l$ and our constant linear decay schedule.**
>
> From the table, we find that, in deeper layers, $\alpha_l$ values decrease and become unstable and the feature quality is even worse than our simpler linear schedule. The unstable nature of learnable $\alpha_l$ might raise a concern for deeper models. This suggests that $\alpha$ is more appropriately regarded as a regularization hyperparameter rather than a learned component of the model.
>
> ## 4. Comparison to stochastic depth
> Training with stochastic depth, as suggested by Reviewer Hwrn, achieves the opposite of our desired effect. Stochastic depth randomly drops a parameterized layer, keeping the residual connection that runs parallel to it. This increases the relative contribution of the residual link to downstream layer inputs, thus amplifying the importance of that residual connection. We want to decay, not amplify, residual connection strength.
>
> To verify that dropping layers, as in stochastic depth, is not a replacement for decaying identity shortcuts, we perform the following experiment. We keep the same training configurations for our ImageNet-100 MAE experiments and replace our proposed decayed identity shortcuts with randomly dropping network blocks as implemented in [1]. Following [1], we use the same linear decay scheduler, and layer $l$ is dropped with a chance of $\frac{l}{L}\cdot \text{Drop}_{\text{min}}$ during training. We report the linear probing of final representations as follows:
>
> |$\text{Drop}_{\text{min}}$|0.0 (baseline)|0.1|0.2|0.3|0.4|
> |:---:|:---:|:---:|:---:|:---:|:---:|
> |Probing Acc|**76.5**|76.4|76.3|74.7|73.5
>
> **Table R7: Comparison to stochastic depth.**
>
> The more we increase the drop chance, the more stochastic depth harms generative representation learning.
>
> ## 5. Experiments with BEiT
> We apply our method to BEiT in adherence to the original configuration, with the addition of skip connections between encoder and decoder, and decaying the encoder's residual connections with our formulation. We run experiments on ImageNet-100 and report the highest linear probing accuracy across all layers, demonstrating enhanced performance over baseline.
>
> |$\alpha_{\text{min}}$ = 0.8|$\alpha_{\text{min}}$ = 1.0(Baseline)|
> |:---:|:---:|
> **74.2**|71.8
>
> **Table R8: Results with BEiT.** Our method improves the probing accuracy compared to the baseline with the full residual connection.
>
> ## 6. Qualitative Comparison of Generated Images
>
> We have updated our supplementary material and Appendix Figure 9 provides visualization of generated images from the conditional diffusion models, comparing our method with $\alpha_{\text{min}} = 0.6$ to the baseline $\alpha_{\text{min}} = 1.0$.
>
> [1] Huang et al., Deep Networks with Stochastic Depth. ECCV, 2016.

---

### Author Response · Authors · 2024-12-04
**Post Rebuttal Reply**

Dear Area Chairs and Reviewers,

We want to thank you again for your feedback!

**1. Primary Results**

As we approach the final day of the rebuttal phase, we would like to provide a summary of our primary results
(including the new ones added during rebuttal), to facilitate a comprehensive evaluation of our work.

|Method|Model|Dataset|Metric|Baseline|Ours|
|:---:|:---:|:---:|:---:|:---:|:---:|
|MAE|ViT-B|ImageNet-1K|KNN(K=20)|27.4|**63.9**|
|MAE|ViT-B|ImageNet-1K|Linear Probing|67.8|**72.7**|
|MAE|ViT-L|ImageNet-100|Linear Probing|81.4|**86.0**|
|MAE|ViT-B|ImageNet-100|Linear Probing|76.5|**83.6**|
|MAE|ViT-S|ImageNet-100|Linear Probing|69.2|**78.5**|
|Diffusion|ViT-B|ImageNet-100|Linear Probing|72.8|**76.1**|
|Diffusion|ViT-B|ImageNet-100|FID|6.93|**4.98**|
|Diffusion|CNN|CIFAR-100|Linear Probing|62.5|**66.9**|
|Diffusion|CNN|CIFAR-100|FID|14.34|**8.99**|
|BEiT|ViT-B|ImageNet-100|Linear Probing|71.8|**74.2**|

**Table RS: Comparison to the baseline with the full residual connection.** Across various models and setups,
our method consistently outperforms the baseline, demonstrating significant improvements in (1) linear probing accuracy, (2) KNN accuracy, and (3) FID score.

**2. Scalability and Selection of Optimal $\alpha_{\text min}$**

We have shown the scalability and robustness of our method through extensive experiments, over a broader range of model configurations (as outlined in the General reply).  The optimal value of our hyper-parameter $\alpha_{\text min}$ changes in a consistent manner with network depth, where the effective decay rate (cumulative decay over the entire network) is key and can be generalized to other model architectures in the same class.

---

### Meta-Review · Area_Chair_z3ti · 2024-12-22

**Metareview:**

The paper proposes two architectural changes to ViT that can greatly improve MAE's linear probing performance. The motivation and intuition behind the modifications (e.g., how residual connections might harm abstract feature learning) are clear and align with hierarchical representation principles. The method is easy to implement (a decaying shortcut weight) and potentially adaptable for large-scale training. Weaknesses of the paper include its limited scope, as experiments focus narrowly on MAE and diffusion models, without demonstrating generalization to other self-supervised techniques or broader architectures. The method underperforms in fine-tuning, which is crucial for MAE’s strengths, and fails to achieve competitive linear probing results compared to SOTA methods like DINOv2. Additionally, the approach introduces sensitivity to hyperparameter choices with no systematic methodology for selection, making practical adoption challenging. The results also raise questions about robustness and scalability across different datasets and model depths.

The work remained borderline after rebuttal phase, but given the original claims and conclusions (e.g. covering generative representation learning) the AC feels that the main claims should have been fully substantiated for acceptance.

**Additional Comments On Reviewer Discussion:**

I would liked a bit more discussion from the reviewers during this phase, but they did raise a few key concerns that were not addressed by the authors.

---

### Decision · Program_Chairs · 2025-01-22

Reject